# Plant functional traits determine latitudinal variations in soil microbial function: evidence from forests in China

Zhiwei Xu[1,2,3], Guirui Yu[4,5,*], Qiufeng Wang[4,5,*], Xinyu Zhang[4,5], Ruili Wang[6], Ning Zhao[7], Nianpeng He[3,4], Ziping Liu[1,2,3]

1. Key Laboratory of Geographical Processes and Ecological Security of Changbai Mountains, Ministry of Education; School of Geographical Sciences, Northeast Normal University, Changchun, 130024, China

2. Institute for Peat and Mire Research, Northeast Normal University, Changchun, 130024, China

3. Jilin Provincial Key Laboratory for Wetland Ecological Processes and Environmental Change in the Changbai Mountains, Changchun, 130024, China

4. Key Laboratory of Ecosystem Network Observation and Modeling, Institute of Geographic Sciences and Natural Resources Research, Chinese Academy of Sciences, Beijing 10010, China.

5. College of Resources and Environment, University of Chinese Academy of Sciences, Beijing, 100190, China

6. College of Forestry, Northwest A&F University, Yangling, 712100, China

7. Key Laboratory of Remote Sensing of Gansu Province, Heihe Remote Sensing Experimental Research Station, Cold and Arid Regions Environmental and Engineering Research Institute, Chinese Academy of Sciences, Lanzhou 730000, China

* Corresponding author at: Key Laboratory of Ecosystem Network Observation and Modeling, Institute of Geographic Sciences and Natural Resources Research, Chinese Academy of Sciences, Beijing 100101, China.No. 11A, Datun Road, Chaoyang District, Beijing, 100101, China. Tel.: +86-10-64889268; fax: +86 10 64889432.

E-mail: yugr@igsnrr.ac.cn (G. Y.), qfwang@igsnrr.ac.cn (Q. W.)

**Abstract.** Plant functional traits have increasingly been studied as determinants of ecosystem
properties, especially for soil biogeochemical processes. While the relationships between biological
community structures and ecological functions are a central issue in ecological theory, these
relationships remain poorly understood at the large scale. We selected nine forests along the North–
South Transect of Eastern China (NSTEC) to determine how plant functional traits influence the
latitudinal pattern of soil microbial functions, and how soil microbial communities and functions are
linked at the regional scale. We found that there was considerable latitudinal variation in the profiles
of different substrate use along the NSTEC. Specifically, we found that the substrate use by
microorganisms was highest in the temperate forest soils (soil microbial substrate use intensities of
10–12), followed by the subtropical forest soils (soil microbial substrate use intensities of 7–10), and
was least in the coniferous forest soils (soil microbial substrate use intensities of 4–7). The latitudinal
variation in soil microbial function was more closely related to plant functional traits (leaf dry matter
content, leaf C concentrations, and leaf N concentrations, $P=0.002$) than climate (Mean annual
precipitation, $P=0.022$). The soil silt, leaf dry matter, and leaf C and N contents were the main
controls on the biogeographical patterns of microbial substrate use in these forest soils. The soil
microbial community structures and functions were significantly correlated along the NSTEC. Soil
carbohydrate and polymer substrate use were mainly related to soil $G^+$ bacterial and actinomycic
PLFAs, while the use of amine and miscellaneous substrates were related to soil $G^-$ bacterial and
fungal PLFAs. The enzyme production varied with changes in the soil microbial communities. The
soil enzyme activities were positively correlated with the bacterial PLFAs but were not correlated
with the fungal PLFAs. The soil organic matter (SOM) decomposition rates were significantly higher
in the temperate forests than in the subtropical and tropical forests, emphasizing the rapid
degradability of high-energy substrates, such as soil microbial biomass carbon, carbohydrates, and
amino acids. The SOM decomposition rates were significantly and negatively related to soil dissolved
organic carbon concentrations, carboxylic acids, polymers, and miscellaneous substrate use. The
relationships between soil PLFAs and microbial substrate use, enzyme activities, and SOM
decomposition rate show that, as the soil microbial community structure changes, soil
biogeochemical processes also change.

**Abstract**

**Abbreviations**
NSTEC      North-South Transect of Eastern China
AWCD      Average well color development
RDA      Redundancy analysis
Soil microbial community
PLFAs      Phospholipid fatty-acids
$G^+$      Gram positive bacteria
$G^-$      Gram negative bacteria
F/B      Fungi/Bacteria
Climate conditions
MAT: Mean annual temperature
MAP: Mean annual precipitation
Soil enzyme activities
BG      β-glucosidase
NAG      N-acetylglucosaminidase
AP      Acid phosphatase
LAP      Leucine aminopeptidase
Soil properties
SMC      Soil moisture content
SOM      Soil organic matter
SOC      Soil organic carbon
TN      Total Nitrogen
DOC      Dissolved organic carbon
MBC      Microbial biomass carbon
Silt      Soil silt fractions (<53 μm)
Plant functional properties:
CWM      Community-weighted means
SLA      The specific leaf area
LDMC      Leaf dry matter content
Leaf C      Leaf C concentrations
Leaf N      Leaf N concentrations

## 1 Introduction

The catabolic diversity of soil microbial communities is a useful indicator of how microbial functions adapt to environmental stress. It can be used to test fundamental questions about soil biological resistance and resilience (Jagadamma et al., 2014; Swallow and Quideau, 2015), and help us understand the role of microbial communities in different environments (Preston-Mafham et al., 2002). Biological community structure and function are intimately linked in ecological processes, and their relationships are a central issue in ecological theory (Talbot et al., 2014). Therefore, a major goal in ecological research is to identify and understand the mechanisms and relationships that control the structure and function of microbial communities over large spatial scales.

Numerous studies have documented how environmental and anthropogenic perturbations impact on the structure, diversity (Tu et al., 2016; Zhou et al., 2016), and enzyme activities (Peng and Wang, 2016; Xu et al., 2017) of soil microbial communities, and have reported that forests in the same climatic zone develop similar microbial communities. Other researchers have examined spatial patterns in soil microbial function at different scales. For example, Tian et al. (2015), from their study of Changbai Mountain, China, found that the soil microbial metabolic activity and functional diversity were spatially dependent. Others reported that soil microbial activities varied by forest type, with high local variation and significant separation along regional climate gradients (Brockett et al., 2012; Cao et al., 2016). Soil microbes from different climatic zones have different affinities for carbon substrates. For example, microorganisms from boreal pine forest soils used carboxylic acids more efficiently, but decomposed amino acids much less efficiently, than microorganisms from temperate forest soils (Klimek et al., 2016). The soil microbial metabolic abilities are also influenced by the dominant tree species, through the production of chemically-unique litter and root exudates, and the soil physico-chemical properties (Menyailo et al., 2002). Despite this, because of limitations in analytical methods, questions still remain about how soil microbial functions vary at the regional scale.

The functional diversity of soil microbial communities is regulated by physico-chemical soil properties (Gartzia-Bengoetxea et al., 2016), climate (Cao et al., 2016), and the composition of plant cover (Sherman and Steinberger, 2012). For example, the geographic patterns in soil microbial

activities mainly reflect the climate, soil pH, and total phosphorus concentrations over large
geographic scales (Cao et al., 2016). Research has shown that substrate-induced respiration rates
were higher in soil microbial communities that developed under beech and holm oak forests than
under oak and pine forests (Gartzia-Bengoetxea et al., 2016). Plant functional traits have increasingly
been studied as determinants of ecosystem properties, especially for soil biogeochemical processes
(De Vries et al., 2012; Pei et al., 2016). Soil bacteria phospholipid fatty-acids (PLFAs) were found
to be positively correlated with the community-weighted means (CWM) of plant functional traits
(leaf nitrogen (N) concentration) (De Vries et al., 2012). The plant leaf dry matter content and the
leaf carbon (C) to nitrogen (N) ratio both influence the multivariate soil microbial community
structure, and these factors positively promote the abundances of specific microbial functional groups
(Pei et al., 2016). Limited soil resources, particularly in tropical forests, mean that soil
microorganisms may be more reliant on plants than soil for C and nutrients via rhizosphere exudation
or litter production, which varies among plant species (Russell et al., 2007; Raich et al., 2014; Waring
et al., 2015). While soil functional diversity has been used as an indicator of microbial metabolic
potential, there have been few studies of the integrated effects of climate, vegetation, and soil
substrate availability on large-scale soil microbial functional diversity.
Although the functional characteristics of soil microorganisms are at least as important as their
patterns of community structure in biogeochemical studies, the links between microbial community
structure and microbial functions are poorly understood. There are two current hypotheses about how
microbes determine ecosystem process rates. In functional redundancy, different microbes perform
the same function and so changes in the microbial community structure do not necessarily lead to a
change in soil function (Balser and Firestone, 2005; Strickland et al., 2009). For example, Banerjee
et al. (2016) showed that the abundance of different bacterial and fungal groups changed by up to
300-fold under straw- and nutrient-amended treatments but that the decomposition rate remained
similar, indicating possible functional redundancy. The functional redundancy hypothesis has
recently been challenged by a counter-hypothesis, referred to as functional dissimilarity, which
suggests that diversity brings stability, and that every species plays a unique role in ecosystem
function (Fierer et al., 2007; Waldrop and Firestone, 2006). Soil microbial community composition
therefore, combined with environmental variables, may ultimately determine ecosystem process rates.
Waldrop and Firestone (2006) showed that gram positive bacteria (G$^+$) were mainly responsible for
the decomposition of pine needles and soil organic matter, but gram negative bacteria (G$^-$) were
mainly responsible for the decomposition of starch and xylose, which are easy to break down.
Philippot et al. (2013), when studying the diversity of denitrifiers, showed that the loss of microbial
diversity could result in decreases of between 4- and 5-fold in denitrification activity. In the
Mediterranean, losses in the mass of decomposing leaf litter from shrub species accelerated as
detritivore assemblages became more functionally dissimilar (Coulis et al., 2015). Research to date
suggests that the different microbial communities will result in variations in soil microbial function
and soil biochemical processes, so information about the relationships between soil microbial
communities and their functions in natural ecosystems is urgently needed.
The North-South Transect of Eastern China (NSTEC) extends from a cold temperate coniferous
forest in the north to a tropical rainforest in the south, and includes almost all the forest types found
in the Northern Hemisphere (Zhang and Yang, 1995) (Fig. 1 and Table 1). This transect, therefore,
provides the optimal environment for investigating large-scale geographical patterns in microbial
communities and their responses to environmental changes. In this study, we examined spatial
patterns in soil labile C concentrations, soil organic matter (SOM) decomposition rates, and
metabolic activity and functional diversity of microbes in nine forest biomes along the NSTEC. We
assessed how abiotic factors, such as climate, soil physical and chemical properties, and biotic factors,
in the form of community-weighted means (CWM) of plant functional traits, contributed to soil
functional diversity at the regional scale. We also examined the links between soil microbial
community structure (PLFAs) and function (SOM decomposition rate, enzyme activities, and
microbial substrate use). We tested four hypotheses in this study, as follows: (1) The profiles of soil
microbial substrate use vary along a latitudinal gradient, (2) biogeographical patterns of soil
microbial substrate use are constrained by climate and plant functional traits, and (3) different soil
microbial communities may have substrate use profiles and SOM decomposition rates.
**2 Material and methods**
2.1 Study area and soil sampling
We selected nine forest ecosystems along the NSTEC, namely Huzhong (HZ), Liangshui (LS),
Changbai (CB), Dongling (DL), Taiyue (TY), Shennong (SN), Jiulian (JL), Dinghu (DH), and
Jianfeng (JF) (18°44′–51°46′N, 128°53′–108°51′E) (Fig. 1, Table 1). Further information about the
soil types and sites has been documented previously by Xu et al. (2017). Forest soils have been
classified following the U.S. soil taxonomy and are described in Table 1 (Soil Survey Staff, 2010),
where information about the climate and the dominant vegetation at each site is also presented.
Soil samples were collected from four random plots at each site in July and August 2013, as
described previously by Xu et al. (2017). Briefly, we established four sampling plots measured $30 \times$
40 m and collected soil samples from a depth of between 0 and 10 cm at between 30 and 50 points
in each plot along an S-shape. On return to the laboratory, the fresh soil samples were immediately
sieved through a 2-mm mesh and subdivided into three subsamples. One subsample was stored briefly
at 4 °C until analysis for soil enzyme activities and soil pH. Another was stored briefly at −20 °C
until analysis for PLFAs and Eco-Biolog. The third was air-dried, sieved through a 0.25 mm mesh,
and analyzed for soil nutrients.
2.2 Soil chemical analyses
Soil pH was measured at a soil-to-water ratio of 1:2.5. The soil moisture content (SMC) was
measured gravimetrically on 20 g fresh soil that was oven-dried at 105 °C to constant weight
immediately on arrival at the study sites' laboratories (Liu et al., 2012a). Soil organic carbon (SOC)
and total N (TN) concentrations were determined by dry combustion of ground samples (100-mesh)
in a C/N analyzer (Elementar, Vario Max CN, Germany). Total phosphorus (TP) was determined
with a flow injection auto-analyzer following digestion with $H_2SO_4$-$HClO_4$ (Huang et al., 2011).
After extraction with distilled water at a soil:distilled water ratio of 1:5, dissolved organic carbon
(DOC) concentrations were determined by Liqui TOC II (Elementar, Liqui TOC II, Germany) (Jones
and Willett, 2006). Soil microbial biomass carbon (MBC) was measured using the chloroform
fumigation and direct extraction technique (Vance et al., 1987). A conversion factor of 2.64 was used
to convert extracted C to biomass C. The silt fractions (<53 μm) of the samples were separated by
wet-sieving and then were freeze-dried in the laboratory, as described by Six et al. (2000). The soil
properties are shown in Table 2. We followed the method described by Bååth et al. (2003) for PLFA
analysis and PLFAs are expressed in units of nmol g$^{-1}$. The four enzymatic activities of β-glucosidase
(BG), N-acetylglucosaminidase (NAG), acid phosphatase (AP), and leucine aminopeptidase (LAP)
responsible for soil C, N, and phosphorous cycling, were measured following the procedure outlined
in Saiya-Cork et al. (2002) and are expressed in units of nmol h$^{-1}$ g$^{-1}$. Information about PLFA and
enzyme activities are presented in Table S1.
The Biolog-ECO plates were purchased from Biolog, US. The substrates for BG, NAG, AP, and
LAP were 4-MUB-β-D-glucoside, 4-MUB-N-acetyl-b-D-glucosaminide, 4-MUB-phosphate, and L-
Leucine-7-amino-4-methylcoumarin, and were stored at −20 °C. An MUB standard was used for the
BG, NAG, and AP enzymes and an AMC standard was used for the LAP enzyme. The substrates and
standards were purchased Sigma. Analytical grade reagents were used for the soil nutrient analysis.
2.3 Vegetation data
We established four sampling plots (30×40 m) in each forest ecosystem. In each plot, we recorded all
the tree individuals, and measured the height and diameter-at-breast-height (DBH) of each woody
individual with a DBH≥2 cm. The diversity of the tree species in the sampling plots was represented
by H′, and the diversity (H′, Shannon-Wiener) of the tree species in the community was calculated
as follows:
$$H' = \sum_{i=0}^{n}(Pi\,lnPi)$$

Where $P_i$ was the importance value of the species $i$ as a proportion of all species, and $n$ was the
number of the species.
We also calculated the community-weighted means (CWM) values of the tree traits using the
cover of each tree. As described by Xu et al. (2018), we collected litter and sun-exposed and mature
leaves (leaf blades for trees) from between five and ten individuals of each plant species at each site
and determined their TN and TC concentrations. We calculated the specific leaf area (SLA, the one-
sided area of a fresh leaf divided by its oven-dried mass, m$^2$ kg$^{-1}$), leaf dry matter content (LDMC,
the oven-dried mass of a leaf divided by its water-saturated fresh mass, mg g$^{-1}$), leaf C concentrations
(leaf C, g kg$^{-1}$), and leaf N concentrations (leaf N, g kg$^{-1}$) for ten fully expanded leaves of each
sampled individual. To measure the leaf traits at the community level, we calculated the CWM of the
tree layer, as follows:

$$\text{CWM} = \sum_{i=1}^{n} pi \times \text{trait}_i$$

Where *pi* is the relative contribution of the species *i* to the cover of the whole community, *n* is the
number of the most abundant species, and trait *i* is the trait value of species *i*, as described by Garnier
*et al*. (2004). The diversity of the tree species and plant functional traits are summarized in Table S2.
2.4 Microbial substrate use
Microbial functional diversities were determined using a Biolog EcoPlate™ (Biolog Inc., Hayward,
California, USA) as described by Garland and Mills (1991). Briefly, approximately 10 g of fresh soil
was suspended in 100 ml saline solution (0.85% NaCl) and shaken on an orbital shaker for 30 min at
190 rpm. A 150 μl aliquot of supernatant from 1:1 000 dilutions of each soil sample was added to
each well. The plates were incubated at 25°C, and the absorbance at 590 nm was measured using a
microplate reader (GENios Pro™, Tecan Trading AG, Männedorf, Switzerland) every 24 h up to 240
h (0, 24, 48, 72, 96, 20, 144, 168, 192, 216, and 240 h). To minimize the influence of cell density in
comparisons among samples, results can be analyzed at constant average well color development
(AWCD). The AWCD for each microplate was calculated by subtracting the control well optical
density (OD) from the substrate well OD (blanked substrate wells), setting any resultant blanked
substrate wells with negative values to 0 and taking the mean of the 95 blanked substrate wells.

243       The Richness (*R*), Shannon-Weiner diversity index (*H'*), Shannon evenness index (*E*), and

Simpson dominance index (*D*) were calculated from the absorption values after EcoPlate™
incubation for 96 h (Gomez et al., 2006). Additionally, the 31 C sources were divided into six groups,
namely carbohydrates, carboxylic acids, amines, amino acids, polymers, and miscellaneous, as
suggested by Zak et al. (1994). The average absorbance of all C sources within each group was
computed as the intensity of the single substrate use. The soil total microbial metabolic intensities (*S*)
of six carbon sources were estimated by the area underneath *AWCD* vs. *t*, and were obtained by
integrating the equation against time t (Guckert et al., 1996):

$$S = \sum [(v_i + v_{i-1})/2 \times (t_i + t_{i-1})]$$

Where $v_i$ was the average optical density of the *i*th incubation time.
2.5 SOM decomposition rate
Four replicates from each sampling site with a 60% water-holding capacity were incubated at 20°C.
In brief, 40 g of each fresh soil sample were put into a 150-ml incubation bottle, and the samples
were then adjusted so that their moisture content corresponded to a water-holding capacity of 60%.
During the 4-week incubation period, the soil respiration rates were measured on days 1, 7, 14, 21,
and 28 using an automatic system. The SOM decomposition rates were calculated as described in the
study of Xu et al. (2015).
2.6 Statistical analysis
One-way analysis of variance (ANOVA) followed by a post hoc Tukey HSD test were used to test
the significance of the differences among the soil properties, C use, functional diversity, and SOM
decomposition rates in the different forest ecosystems. We tested the relationships between labile C,
soil microbial community structure, microbial function, and the SOM decomposition rates with the
Pearson correlation test. Differences were considered significant when $P<0.05$, with the marginal
significance set at $P<0.01$. All P values were adjusted using the Bonferroni correction to account for
multiple comparisons.
We used redundancy analysis (RDA) to examine the relationship between the environmental
variables and soil microbial substrate use. The environmental variables were the same as those
described in Xu et al. (2018), including climate, soil properties, litter properties, and plant functional
traits. Before RDA, we conducted forward selection of the environmental variables that were
significantly correlated with variations in the microbial substrate use profile using stepwise
regression and the Monte Carlo Permutation Test. We used CANOCO software 4.5 (Ter Braak and
Smilauer 2002) for the RDA and stepwise regression. The environmental properties, which were
significantly correlated with the microbial substrate use in the RDA, were stressed in the plots. Path
analysis was conducted to examine the direct and indirect effect of biotic and abiotic factors on soil
microbial use of carbon sources. All ANOVA, regression analyses, and path analysis were performed
using SPSS 19.0 for Windows. Data are reported as the mean ± SE.
**3 Results**
3.1 Patterns in the microbial substrate use, soil labile carbon, and SOM decomposition rates
There was no obvious latitudinal pattern for the soil total microbial metabolic intensity (Fig.2). Of
the forests along the NSTEC, the C metabolic intensity of soil microbes was lowest in HZ and LS;
the C metabolic intensity of soil microbes differed significantly between JF and the other forests (Fig.
2), which indicates that the color development was significantly higher in the tropical forest soils
than in the subtropical and temperate forest soils and is consistent with the variations in the AWCD
(Fig. S1). The average values of $R$, $H'$, and $D$ were significantly different among the nine forest soils
and were highest in JF, SN, and CB (Table 3). Generally, the soil microbial use of the six different
carbon sources were relatively higher in tropical and subtropical climatic area (low latitude) than
those in temperate climatic area (high latitude) (Fig.3).
Across the nine forests, soil microorganisms used the six substrate groups in the same order; the
carboxylic acid substrate was used most, followed by amino acids, carbohydrates, polymers, amines,
and miscellaneous substrates (Fig. 3). Microorganisms in the boreal and temperate forests mainly
metabolized carbohydrates, amino acids, and carboxylic acids, while those from the subtropical and
tropical forests used the substrates in equal proportions. The substrate microbial use ability was
highest in the coniferous broad-leaved mixed forest and tropical forest soils, and lowest in the
coniferous forest soil (Fig. 3).
Overall, soil MBC concentrations in the boreal and temperate forests were three to eight times
higher than those of the subtropical and tropical forests. In contrast, the average DOC concentrations
in the tropical and subtropical forest soils, which ranged from 311 to 458 mg kg$^{-1}$, were significantly
higher than the average concentrations in the temperate and boreal forest soils, where the average
concentrations ranged from 204 to 284 mg kg$^{-1}$ (Table 2). The average SOM decomposition rates in
the subtropical forests ranged from 0.64 to 2.42 μg C g$^{-1}$ d$^{-1}$, and were significantly lower than the
rates in the temperate forests, which ranged from 3.43 to 4.61 μg C g$^{-1}$ d$^{-1}$ (Table S3).
3.2 Effect of environmental properties on soil microbial substrate use
Redundancy analysis showed that the variations in soil microbial substrate use were strongly and
positively correlated with the CWM values of LDMC, leaf N, and leaf C, and strongly and negatively

correlated with the soil silt content and SMC (Fig. 4). The RDA2 of soil microbial substrate use was strongly positively correlated with TN and SOC, but negatively correlated with the mean annual precipitation (MAP) (Fig. 5). RDA1 mainly represented the plant functional traits, soil texture, and micro-meteorological conditions, while RDA2 represented climate and soil nutrients. Overall, the soil silt content and the CWM values of plant functional traits were the main predictors of the latitudinal variation in the soil microbial substrate use along the NSTEC.

3.3 Relationships between soil microbial substrate use, enzyme activities, and PLFAs

Microbial carbohydrate use was positively related with bacterial biomass and actinomycic biomass (Fig. 5). Microbial polymer use was negatively related with bacterial biomass and actinomycic biomass. Microbial amines use was negatively related with $G^-$ bacterial and fungal biomass. Miscellaneous substrate use was positively related with fungal biomass and $G^+/G^-$ bacterial biomass (Fig. 5).

The abundance of $G^-$ bacteria was positively associated first with the specific activities of BG, whereas actinomycetes and $G^+$ bacteria were positively associated with BG and LAP. Soil fungi were negatively associated with BG (Fig. 5).

3.4 Relationships between SOM decomposition rate, PLFAs, enzyme activity, and microbial metabolic activities

The SOM decomposition rates were significantly and positively related to soil MBC concentrations but significantly and negatively related to soil DOC concentrations (Fig. 6a and b). Except for amino acid and amine substrates, the SOM decomposition rates were significantly and positively related to microbial metabolic activities (AWCD) and carbohydrate substrate use (Fig. 6c and d) and negatively related to carboxylic acid, polymer, and miscellaneous substrate use (Fig. 6e, g, and i).

The SOM decomposition rates were significantly and positively correlated with total PLFAs ($r=0.456$, $P=0.005$), bacteria ($r=0.3836$, $P=0.021$), actinomycetes ($r=0.500$, $P=0.002$), and $G^-$ bacteria PLFAs ($r=0.520$, $P=0.001$) (Fig. 7a, b, d, and f) but were negatively correlated with fungal PLFAs ($r=-0.370$, $P=0.026$), F/B ($r=-0.513$, $P=0.001$), and the $G^+/G^-$ ($r=-0.496$, $P=0.002$) (Fig. 7c, g, and h). Except for LAP activity, soil enzyme activities were significantly and positively correlated

with the SOM decomposition rates (*P*<0.01) (Fig. 7i, j, and l).
**4 Discussion**
4.1 Response of soil labile C and SOM decomposition rates to variations in forest type
Soil organic matter is one of the most important C pools in terrestrial ecosystems. The concentrations
of soil DOC in the temperate forests were lower than those in subtropical forests but the soil MBC
concentrations were higher in temperate forests than in subtropical forests. This reflects the results
of previous regional and global studies (Tian et al., 2010; Xu et al., 2013), and shows that the
production/consumption ratio of soil DOC was lower, but that microbial C immobilization was higher,
in the high latitude forests than closer to the tropics (Fang et al., 2014). Soil DOC, as a labile SOM
fraction with a rapid turnover, is one of the primary energy sources for microorganisms. The higher
temperatures and precipitation in subtropical and tropical forests lead to higher turnover rates (Fang
et al., 2014), so soil DOC concentrations were highest in subtropical, and MBC concentrations were
lowest, in tropical forests. However, in temperate forests, more C is assimilated into microbial
biomass, so that less C is lost through chemical and physical processes (Liu et al., 2010). Also,
because the decomposition ability of different microbe groups varies, the differences in the soil
microbial communities in different forest ecosystems may also be responsible for the spatial
variations in the soil DOC and MBC concentrations along the NSTEC (Hagedorn et al., 2008).
Heterotrophic soil respiration is sustained by the decomposition of SOM. The SOM
decomposition rates along the NSTEC were greater in temperate forests than in subtropical forests,
which was consistent with the variations in the soil MBC and SOC concentrations. These results
indicate that, as found in other studies, large scale SOM decomposition rates are driven by the
amounts of substrate available (Yu et al., 2010). Changes in the availability of C in SOM may affect
the microbial resource strategies, which may in turn influence the SOM decomposition rate. Some
forest soils were intermittently saturated (such as CB, Table 2) or high with mean annual precipitation.
Under the anaerobic conditions, soil organic decomposition is mediated by a complex suite of
microbial processes (Megonigal et al., 2004). The fermentation products including low molecular
weight alcohols, fatty acids, and dihydrogen can serve as substrates for anaerobic respiration using a
variety of alternative terminal electron acceptors in place of oxygen to mineralize organic carbon to
carbon dioxide ($CO_2$). Therefore, the soil organic matter decomposition rate might be slow in these
anaerobic conditions. These results demonstrate that the reduction of organic matter is a key step of
anaerobic decomposition (Keller and Takagi, 2013).
4.2 Mechanisms driving latitudinal variations in microbial substrate use
The AWCD reflects the sole C source use ability of the soil microbial community (Garland and Mills,
1991). Of the six groups of C substrates, microbial communities in the temperate forests mainly used
carbohydrates, carboxylic acids, and amino acids, which suggests that microorganisms in temperate
forests probably use high-energy substrates that degrade easily (Kunito et al., 2009). The carbon
substrate use was lowest in the boreal coniferous forest (HZ). This shows that, compared with
coniferous species, broadleaved tree species produce root exudates and litter high in water-soluble
sugars, organic acids, and amino acids that are more favorable for microbial activity (Priha et al.

373 2001).

There was no significant latitudinal pattern in the soil total microbial metabolic intensity (S) in
our study, which was inconsistent with hypothesis (1). However, there was significant latitudinal
variation in the use of the six different carbon sources. The soil microbial use of carbon sources were
relatively higher in tropical and subtropical climatic area than those in temperate climatic area.
Consistent with hypothesis (2), soil microbial functions were similar in closely related to the CWM
values of the tree species (Fig. 4). However, only MAP of climatic factors had a moderate effect on
the soil microbial function (Fig. 4). Climate may have indirect effect on the latitudinal pattern of
different carbon sources use. In our study, MAT and MAP affected the six different carbon sources
use indirectly by influencing the soil temperature (ST), soil total nitrogen (TN), and total phosphorus
(TP) (Table S4-9). It was reported that climate was the main environmental parameters driving the
latitudinal patterns of community-level leaf traits through the regulation of species composition
(Wang et al., 2016). Woody plant species with evergreen broadleaves dominated in the tropic regions
characterized by hot, humid, and infertile habitats showed low CWM values of leaf N and SLA and
high CWM values of LDMC (Table S2).
A growing number of studies reported that vegetation type, land use, soil nutrients, and soil
organic matter quality and quantity can determine large scale patterns of microbial communities (de
Vries et al., 2012; Tu et al., 2016). Plant functional traits that are related to growth may determine a
tree species' ability to contribute to the soil carbon pool via leaf litter inputs. For example, it was
previously reported that plant traits such as the leaf N content, SLA, and LDMC could explain
variations in soil nutrients and litter decomposition rates (Eichenberg et al., 2014; Laughlin, 2011).
Therefore, we examined how these plant traits influenced the soil microbial function by latitude. We
found that changes in the soil microbial C substrate use with latitude were mainly related to the soil
silt contents and the CWMs of LDMC, and leaf C and leaf N concentrations, which indicates that the
quality of nutrients from plants had a major influence on microbial carbon use efficiency (Hypothesis
(2)). Plant species with high a SLA, high leaf N concentrations, and low LDMC can produce
bacterial-dominated soil microbial communities in grasslands (Orwin et al., 2010). Looking beyond
individual traits, related tree species may cultivate microbial communities with similar preferences
for carbon sources through coevolution of plants and microbes (Liu et al., 2012b; Buscot, 2015).
As hypothesized, the soil microbial community composition was explained by the CWMs of
plant traits at the regional scale. Carbon substrate use was negatively correlated with the CWM of
leaf N concentrations (Table S2, Fig. S2). Bacterially dominated soil microbial communities develop
from leaf litter comprised of N-rich leaves from fast growing species (De Vries et al., 2012), while
leaves with low N concentrations will promote fungal domination (Orwin et al., 2010; De Vries et
al., 2012). In line with this, fungal biomass decreased, and bacterial biomass increased, as the CWM
leaf N content increased, and is associated with fast-growing, N-exploitative plants (Xu et al., 2018).
Leaf N concentrations are considered as indicators of plant growth and resource uptake (Wright et
al., 2004). The results from this study show that, along the NSTEC, high leaf N restrained microbial
C substrate use (Fig.S2) and was a good indicator of the competition between plants for soil N (Pei
et al., 2016). Soil microbes and nearby plants may have been competing for N in the soil.
We also found that the C substrate use was negatively correlated with the CWM of leaf C
concentrations (Table S2, Fig. S2). Plants at high latitudes may have higher leaf C concentrations
than plants at lower latitudes so that they can balance the osmotic pressure of cells and resist freezing
(Millard et al., 2007; Hoch and Körner, 2012). The increased C was most likely in the form of an
increase in non-structural C, including starch, low molecular weight sugars, and storage lipids that

are easy to break down. Therefore, soil microorganisms from the temperate forests mainly metabolized high-energy substrates, such as carbohydrates, carboxylic acids, and amino acids.

The LDMC is the ratio of the leaf dry weight to the fresh weight and has been used as a proxy for the ratio of structural compounds to assimilatory tissue (mesophyll and epidermis, Van Arendonk and Poorter, 1994). High values of LDMC indicate large amounts of vascular tissue, cellulose, insoluble sugars, and leaf lignin that are difficult to decompose (Poorter and Bergkotte, 1992); C substrates such as carbohydrates, carboxylic acid, and amino acid are, however, easy to decompose (Myers et al., 2001). In line with this, the use of carbohydrate, carboxylic acid, and amino acid substrates was negatively related to the CWMs of the LDMC (Table S2). Pei et al. (2016) reported that the LDMC was an important driver of multivariate soil microbial community structure and $G^-$ bacterial abundance.

Soil texture regulates soil biological processes and so affects the soil microbial community structure (Sessitsch et al., 2001). In the present study, microbial C substrate use was significantly and positively related to the soil silt content. Soil types and textures varied along the NSTEC. Soil texture influences how microbes use organic matter, and has a strong influence on soil moisture, nutrient availability, and retention (Veen and Kuikman, 1990). Fine-textured soils with a higher silt content are known to be more favorable for bacterial growth than soils with a lower silt content because of their greater water-holding capacity and nutrient availability, and because they are better protected from bacterial grazers (Carson et al., 2010). We found that the microbial C substrate use was higher in LS, CB, SN, and JL than in the other forests, reflecting their fine-grained soils and high silt contents, which ranged from 60% to 80%.

4.3 Links between soil microbial community structure and function

The soil microbial community structure and functions were significantly correlated along the NSTEC. Soil carbohydrate and polymer substrate use were mainly related to soil $G^+$ bacterial and actinomycic biomass, but amines and miscellaneous substrates were mainly related to soil $G^-$ bacterial, fungal biomass, and the F/B ratio. Soil bacteria mainly decomposed simple carbohydrates, organic acids, and amino acids, whereas soil fungi mainly decomposed recalcitrant compounds (Myers et al., 2001; Treonis et al., 2004). Consistent with this, soil bacterial PLFAs were positively correlated with the

carbohydrate substrate use and the fungal PLFAs were positively related with miscellaneous substrate
use. The results are similar to those reported by Sterner and Elser (2002), who found that fungi tended
to have higher C/N or C/P ratios while heterotrophic bacteria typically have lower C/N or C/P ratios.
Shifts in microbial community composition may influence enzyme production (DeForest et al.,
2012; Waldrop et al., 2000; Brockett et al., 2012). Different microbial groups require different
amounts of nutrients to construct biomass, or have enzymes that differ in their affinity for nutrients.
We found that the relative abundances of the $G^+$ bacteria and actinomycetes communities were
associated with the specific activities of BG, AP, and LAP), whereas the relative abundance of the
$G^-$ bacteria was correlated with soil NAG activities involved in chitin degradation. In agreement with
our study, numerous other researchers have reported significant correlations between PLFA profiles
and enzyme activities (Waldrop et al., 2000; DeForest et al., 2012; Brockett et al., 2012; Riah-Anglet
et al., 2015). Soil BG was mainly responsible for cellulose degradation and was involved in breaking
down complex organic compounds (cellobiose) into small molecule substrates (glucose) in favor of
acquiring C through microbial community growth (Waldrop et al., 2000). Soil NAG activities were
weakly and positively related with fungal biomass in the present study, and may have been mainly
produced by fungal populations (Valášková et al., 2007). Fungi are commonly considered as
producers of oxidative enzymes. Therefore, the influence of fungal biomass on variations in enzyme
activities was minimal (Kivlin and Treseder, 2014). The linkages between enzyme activity and
community composition may provide some insight into the microbial mechanisms that drive the
decomposition of macromolecular C compounds. These results suggest that that overall ecosystem
functioning may suffer if soil microbial groups are lost.
The quality and amounts of SOM are influenced by the biomass, vegetation coverage, root
distribution, and microbial species (Raich and Schlesinger, 1992). The SOM decomposition rates
were higher in temperate forests than in tropical forests and may reflect the higher soil microbial
biomass (Wang et al., 2016). In line with this, SOM decomposition rates were positively related with
soil MBC concentrations and different groups of PLFAs. The inverse relationships between SOM
decomposition rates and DOC, carboxylic acids, polymers, and miscellaneous along the NSTEC,
indicate a shift in the soil C turnover from open to closed with increases in the soil labile C
concentrations (Fang et al., 2014). Soil DOC and MBC influence SOM decomposition rates indirectly
by regulating microbial properties (Boberg et al., 2014; Wei et al., 2014). In our study, SOM
decomposition rates were positively related with bacterial PLFAs but negatively with fungal PLFAs.
Because different communities of microbes have different SOM use efficiencies (Balser and Wixon,
2009; Lipson et al., 2009; Monson et al., 2006), changes in the microbial community structure may
influence the microbial activities and the decomposition rates of organic matter (Lipson et al., 2009;
Keiblinger et al., 2010). The functional dissimilarity of microbes and fungi may help explain these
results. However, we did not measure some key variables, such as the microbial competition and
interactions, and relationship between the microbial diversity and the decomposition rates. Therefore,
in the future, we will use different experimental techniques that will help us gain an improved
understanding of the mechanisms that drive the relationships between the structure and function of
microbial communities.
**5 Conclusions**
In this study, we examined the patterns in labile C concentrations, SOM decomposition rates,
microbial substrate use, and functional diversity and identified a combination of abiotic and biotic
factors that influenced soil microbial functional diversity at the regional scale. The MBC
concentration and SOM decomposition rates were significantly lower, and the soil DOC
concentrations were higher in the subtropical and tropical forests than in the temperate forests. There
was no obvious latitudinal pattern for the soil total microbial metabolic intensity. However, soil
microbial use of the different carbon sources varied with the latitude except the amines carbon source.
Soil microbial use of carbon sources were relatively higher in tropical climatic areas. For the first
time, we showed that CWM values of plant traits explained variations in soil microbial C substrate
use at the regional scale. Additionally, the fine-grained soils with high silt contents were higher in
the microbial C substrate use. Climate factors affected the soil microbial uses of carbon sources
indirectly by influencing the soil temperature (ST) and soil nutrition. Soil microbial community
structure and function were strongly related, which suggests that the loss of soil microbial groups
may have consequences for overall ecosystem functioning.
*Data accessibility*. Requests for data and materials should be addressed to N.H. (henp@igsnrr.ac.cn) and G.Y.
(yugr@igsnrr.ac.cn).

*Author contributions*. Z.W.X., G.R.Y. and X.Y.Z. planned and designed the research. Z.W.X., N.P.H., R.L.W., and
N.Z. conducted fieldwork. Z.W.X., G.R.Y., X.Y.Z., and Q.F.W wrote the manuscript. All authors contributed
critically to the drafts and gave final approval for publication.
*Competing interests.* The authors declare that they have no conflict of interest.

## Acknowledgements

This research was jointly supported by the National Natural Science Foundation of China (41601084, 41571251), the
Fundamental Research Funds for the Central Universities (2412019FZ001), Science and Technology Research Project
of Jilin Province (JJKH20190283KJ), the National Key R&D Program of China (2016YFA0602301), and the China
Postdoctoral Science Foundation (2018M631850).

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

## Figures legends

**Fig. 1.** Distribution of typical forest ecosystems along the North-South Transect of eastern China (NSTEC). The abbreviations for the sampling sites from north to south are as follows: HZ, Huzhong; LS, Liangshui; CB, Changbai; DL, Dongling; TY, Taiyue; SN, Shennong; JL, Jiulian; DH, Dinghu; JF, Jiangfeng. These abbreviations are used for the nine forests throughout.

**Fig.2.** Variations in soil microbial substrate use during a 240-h incubation for the nine forests. Different colors represent different forest types: Yellow, coniferous forest; Dark yellow, coniferous broad-leaved mixed forest; Purple, deciduous broad-leaved forest; Olive, subtropical evergreen broad-leaved forest; Orange, Tropical monsoon forest. Different lowercase letters indicate significant differences among forests in the same climate zone. The abbreviations of the sampling sites are given in Table 1.

**Fig. 3.** Characteristics of microbial use of (a) carbohydrates, (b) carboxylic acids, (c) amino acids, (d) polymers, (e) amines, and (f) miscellaneous along the NSTEC. The representatives of different colors were showed in Figure 2.

**Fig.4.** Redundancy analysis (RDA) ordination biplot of soil microbial carbon sources use efficiency and environmental properties. The representatives of different colors were showed in Figure 2. The dotted lines and solid lines represent the environmental variables and lipid signatures and carbon sources. The abbreviations of the variables in this figure are as follows: MAP, mean annual precipitation. The vegetation data: LDMC, leaf dry matter weight; Leaf C, leaf carbon content; Leaf N, leaf nitrogen content; SLA, specific leaf area. Soil properties included SMC, soil moisture content; Silt, soil silt content; TN, soil total nitrogen; SOC, soil organic carbon. The abbreviations of the sampling sites were given in Table 1

**Fig.5.** The heatmap of the Pearson's correlation coefficients between the use of individual substrates and microbial PLFAs and soil enzyme activities. Note: The abbreviations of the variables: Actino-, actinomycetes; F/B, fungi/bacteria; $G^+$, gram positive bacteria; $G^-$, gram negative bacteria; $G^+/G^-$, Gram-positive bacteria/ Gram-negative bacteria. BG, β-1, 4-glucosidase; NAG, β-1,4-N-acetylglucosaminidase; LAP, leucine aminopeptidase; AP, acid phosphatase. $**P< 0.01$, $*P< 0.05$.

**Fig. 6.** Relationships between soil carbon mineralization rates (μg C $g^{-1}d^{-1}$) and microbial biomass C (MBC), soil dissolved organic C (DOC), average well color development (AWCD), and individual substrate use.

**Fig. 7.** Relationships between soil carbon mineralization rates (μg C $g^{-1} d^{-1}$) and different groups of soil microbial PLFAs (a-h) and enzyme activities (i-l).

## Supporting information

**Table S1** Average values of forest soil enzyme activities and different PLFA groups along the NSTEC.

**Table S2** Plant diversity and community weighted means of plant functional traits

**Table S3** Soil organic matter (SOM) decomposition rates during the28 days of incubation time (Mean±SE) (μg C $g^{-1}d^{-1}$)

**Table S4** Path analysis of environmental variables to soil microbial use of carbohydrates source

**Table S5** Path analysis of relevant environmental variables to soil microbial use of carboxylic acids source

**Table S6** Path analysis of relevant environmental variables to soil microbial use of amino acids source

**Table S7** Path analysis of relevant environmental variables to soil microbial use of polymers source

**Table S8** Path analysis of relevant environmental variables to soil microbial use of amines source

**Table S9** Path analysis of relevant environmental variables to soil microbial use of miscellaneous source

**Fig. S1** Variations in the average well color development (AWCD) values during a 240-h incubation for the nine forests. The abbreviations of the sampling sites are the same as those in Table 1.

**Fig.S2** The Pearson's correlation coefficients between the use of individual substrates and plant functional traits.

**Tables**
**Table 1.** The main characteristics of the sampling sites along the North South Transect of East China

| Sampling Sites | Longitude (E) | Latitude (N) | Elevation (m) | MAT[b] (ºC) | MAP[b] (mm) | Vegetation types | Soil type |
|---|---|---|---|---|---|---|---|
| HZ[a] | 123°01′12″ | 51°46′48″ | 850 | −3.7 | 473 | Cold temperate coniferous forest | Spodosols |
| LS | 128°53′51″ | 47°11′06″ | 401 | 0.01 | 648 | Temperate conifer broad-leaved mixed forest | Albi-Boric Argosols |
| CB | 128°05′27″ | 42°24′16″ | 758 | 2.8 | 691 | Temperate conifer broad-leaved mixed forest | Albi-Boric Argosols |
| DL | 115°25′24″ | 39°57′27″ | 972 | 6.6 | 539 | Warm temperate deciduous broad-leaved forest | Alfisols |
| TY | 112°04′39″ | 36°41′43″ | 1668 | 6.0 | 644 | Warm temperate deciduous broad-leaved forest | Alfisols |
| SN | 110°29′43″ | 31°19′15″ | 1510 | 8.5 | 1447 | Subtropical deciduous evergreen mixed forest | Inceptisols |
| JL | 114°26′28″ | 24°35′05″ | 562 | 18.2 | 1770 | Subtropical evergreen broad-leaved forest | Ultisols |
| DH | 112°32′14″ | 23°10′25″ | 240 | 21.8 | 1927 | Subtropical monsoon evergreen broad-leaved forest | Ultisols |
| JF | 108°51′26″ | 18°44′18″ | 809 | 23.2 | 2266 | Tropical monsoon forest | Ultisols |

a:HZ, Huzhong; LS, Liangshui; CB, Changbai; DL, Dongling; TY,Taiyue; SN, Shennong; JL, Jiulian; DH, Dinghu; JF, Jiangfeng.
b: MAT, mean annual temperature; MAP, mean annual precipitation.
**Table 2.** Soil properties of different sampling sites

| Sampling site | pH | ST (°C) | SMC (%) | Silt (%) | SOC (g kg$^{-1}$) | MBC (mg kg$^{-1}$) | DOC (mg kg$^{-1}$) | TN (g kg$^{-1}$) | TP (g kg$^{-1}$) |
|---|---|---|---|---|---|---|---|---|---|
| HZ | 6.79±0.02a | 10.3±0.15g | 45.3±0.90c | 56±1.2c | 42.29±0.47b | 350±6.0a | 240±7.6e | 2.90±0.16d | 0.87±0.02b |
| LS | 6.17±0.02b | 15.9±0.02f | 46.9±0.76c | 64±0.3b | 62.08±7.20a | 316±0.7a | 204±4.9f | 4.59±0.29b | 0.59±0.02c |
| CB | 6.37±0.04b | 16.0±0.06f | 102.8±0.25a | 76±0.6a | 72.38±2.00a | 178±8.8b | 314±8.6c | 6.05±0.17a | 1.67±0.08a |
| DL | 6.87±0.02a | 17.8±0.14e | 32.4±0.30e | 6±2.4e | 38.83±0.41c | 43±0.8e | 284±2.6d | 3.17±0.04d | 0.56±0.01c |
| TY | 6.85±0.05a | 16.0±0.12f | 36.0±0.23d | 49±1.4d | 41.34±2.75c | 115±4.0c | 226±13.8f | 2.43±0.15e | 0.52±0.01c |
| SN | 6.93±0.01a | 18.4±0.12d | 50.5±0.63b | 74±0.3a | 36.13±1.26c | 72±13.1e | 311±13.2c | 3.76±0.05c | 0.81±0.01b |
| JL | 5.57±0.19b | 25.3±0.01a | 39.0±0.89d | 68±0.3b | 31.55±1.82c | 89±19.7d | 387±1.9b | 2.28±0.09e | 0.36±0.01d |
| DH | 5.43±0.03c | 24.4±0.04b | 37.8±0.38d | 50±1.8d | 28.47±0.54d | 38±0.1e | 334±7.7c | 1.77±0.02f | 0.20±0.01e |
| JF | 6.32±0.01c | 22.5±0.07c | 38.6±0.12d | 49±0.2d | 29.38±0.94d | 140±1.3c | 458±6.6a | 1.99±0.02e | 0.15±0.01e |

Note: ST=temperature of 0–10 cm soil; SMC=soil moisture content; Silt=soil silt content; SOC=soil organic carbon; MBC=microbial biomass carbon; DOC=dissolved organic carbon; TN=soil
total nitrogen; TP=soil total phosphorus. Values were presented as means ± SE (n=4). The abbreviations of the sampling sites were given in the Table 1.
**Table 3.** Functional diversity of soil microbial communities in forest ecosystems along the NSTEC

| Sampling sites | Richness (*R*) | Shannon *H′* | Shannon *E* | Simpson *D* |
|---|---|---|---|---|
| **HZ** | 14.08±0.34d | 2.65±0.03d | 1.01±0.007b | 0.91±0.002c |
| **LS** | 25.29±0.14b | 3.12±0.02b | 0.98±0.003c | 0.95±0.001a |
| **CB** | 27.00±0.27a | 3.22±0.01a | 0.98±0.001c | 0.95±0.001a |
| **DL** | 11.54±0.47e | 2.52±0.03e | 1.04±0.010a | 0.87±0.005d |
| **TY** | 22.33±0.87c | 3.02±0.02c | 0.98±0.002c | 0.94±0.001a |
| **SN** | 28.10±0.34a | 3.24±0.01a | 0.97±0.001c | 0.95±0.001a |
| **JL** | 23.54±0.07c | 3.04±0.01c | 0.96±0.001c | 0.93±0.003b |
| **DH** | 25.65±0.71b | 3.11±0.01b | 0.97±0.001c | 0.93±0.002b |
| **JF** | 27.63±0.68a | 3.19±0.02a | 0.96±0.001c | 0.95±0.002a |

Indices were calculated based on the optical density values after incubation for 96 h. Data are expressed as
means±standard errors. Different lowercase letters indicate significant differences among forests. The abbreviations
of the sampling sites are the same as those used in Table 1.
**Figures**

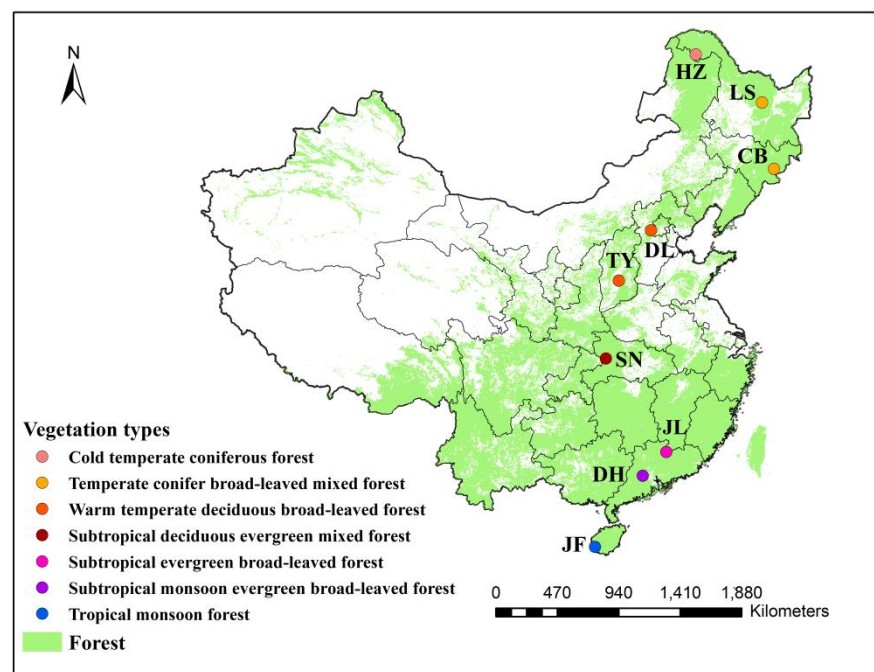

**Figure 1.** Distribution of typical forest ecosystems along the North-South Transect of eastern China (NSTEC).
The abbreviations of sampling sites from north to south are as follows: HZ, Huzhong; LS, Liangshui; CB,
Changbai; DL, Dongling; TY, Taiyue; SN, Shennong; JL, Jiulian; DH, Dinghu; JF, Jiangfeng.

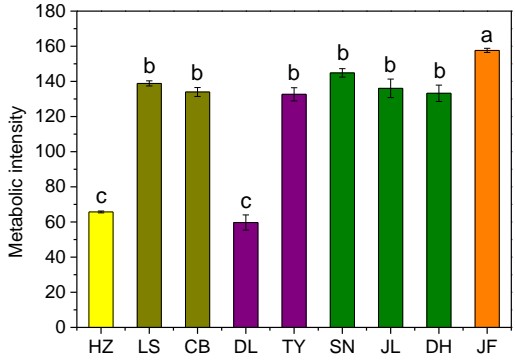

**Figure 2.** Variations in soil total microbial metabolic intensity during a 240-h incubation for the nine forests.
Different colors represent different forest types: Yellow, coniferous forest; Dark yellow, coniferous broad-leaved
mixed forest; Purple, deciduous broad-leaved forest; Olive, subtropical evergreen broad-leaved forest; Orange,
Tropical monsoon forest. Different lowercase letters indicate significant differences among forests in the same
climate zone. The abbreviations of the sampling sites are given in Table 1.

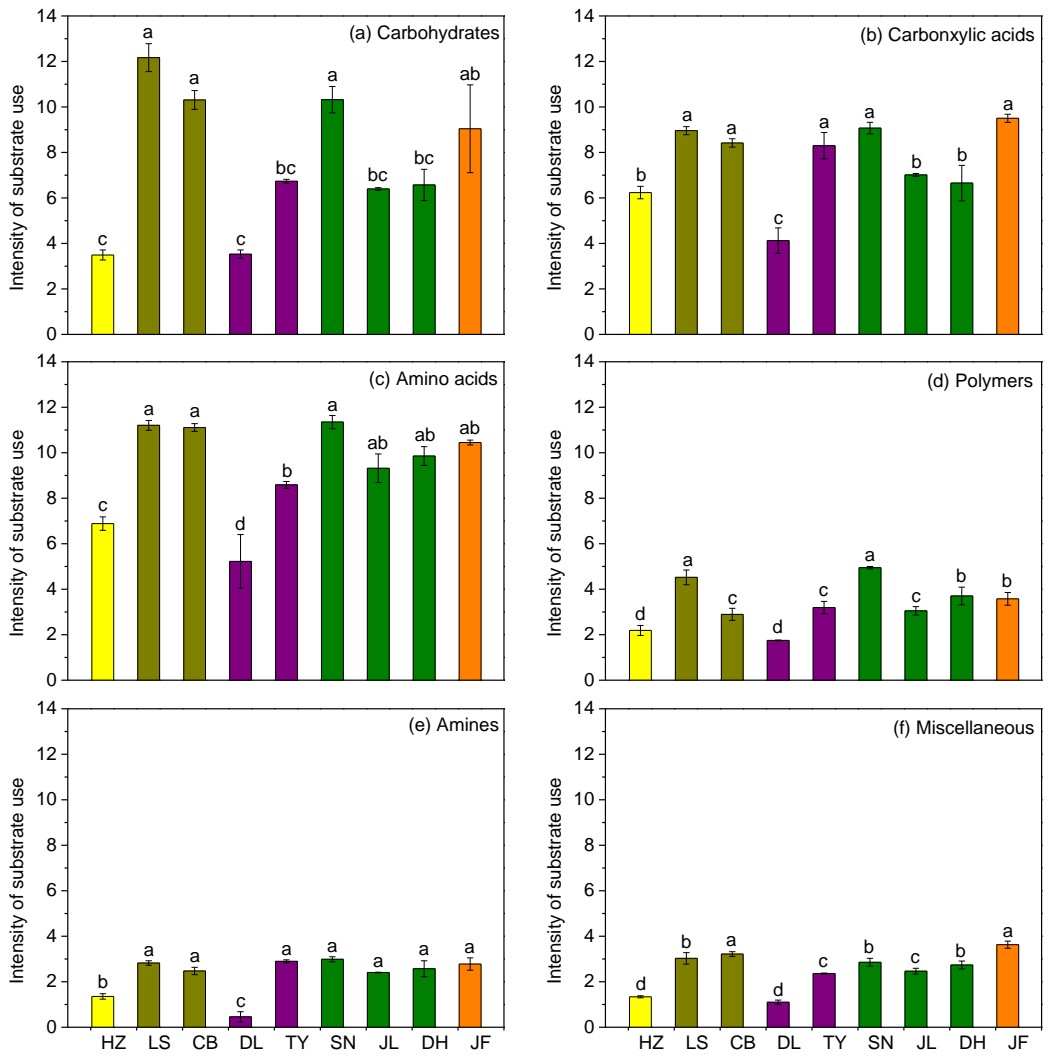

**Figure 3.** Characteristics of microbial use of (a) carbohydrates, (b) carboxylic acids, (c) amino acids, (d) polymers, (e) amines, and (f) miscellaneous along the NSTEC. The representatives of different colors were showed in Figure 2.

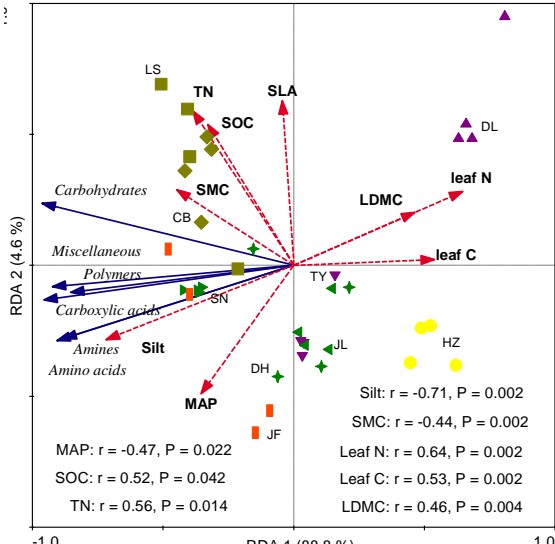

**Figure 4.** Redundancy analysis (RDA) ordination biplot of soil microbial carbon sources use efficiency and
environmental properties. The representatives of different colors were showed in Figure 2. The dotted lines and solid
lines represent the environmental variables and lipid signatures and carbon sources. The abbreviations of the variables
in this figure are as follows: MAP, mean annual precipitation. The vegetation data: LDMC, leaf dry matter weight;
Leaf C, leaf carbon content; Leaf N, leaf nitrogen content; SLA, specific leaf area. Soil properties included SMC, soil
moisture content; Silt, soil silt content; TN, soil total nitrogen; SOC, soil organic carbon. The abbreviations of the
sampling sites were given in Table 1.

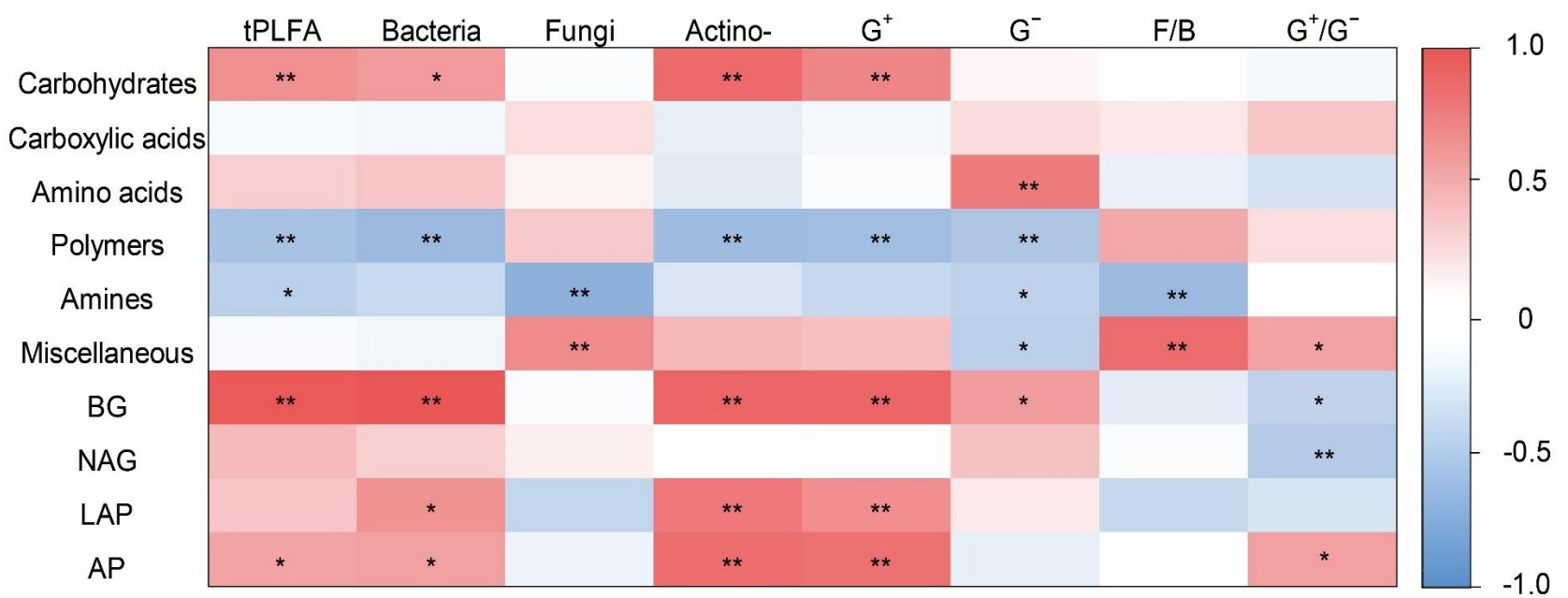

**Figure 5.** The heatmap of the Pearson's correlation coefficients between the use of individual substrates and microbial PLFAs and soil enzyme activities. Note: The abbreviations of the variables:
Actino-, actinomycetes; F/B, fungi/bacteria; $G^+$, gram positive bacteria; $G^-$, gram negative bacteria; $G^+/G^-$, Gram-positive bacteria/ Gram-negative bacteria. BG, β-1, 4-glucosidase; NAG, β-1,4-
N-acetylglucosaminidase; LAP, leucine aminopeptidase; AP, acid phosphatase. **$P< 0.01$, *$P< 0.05$.

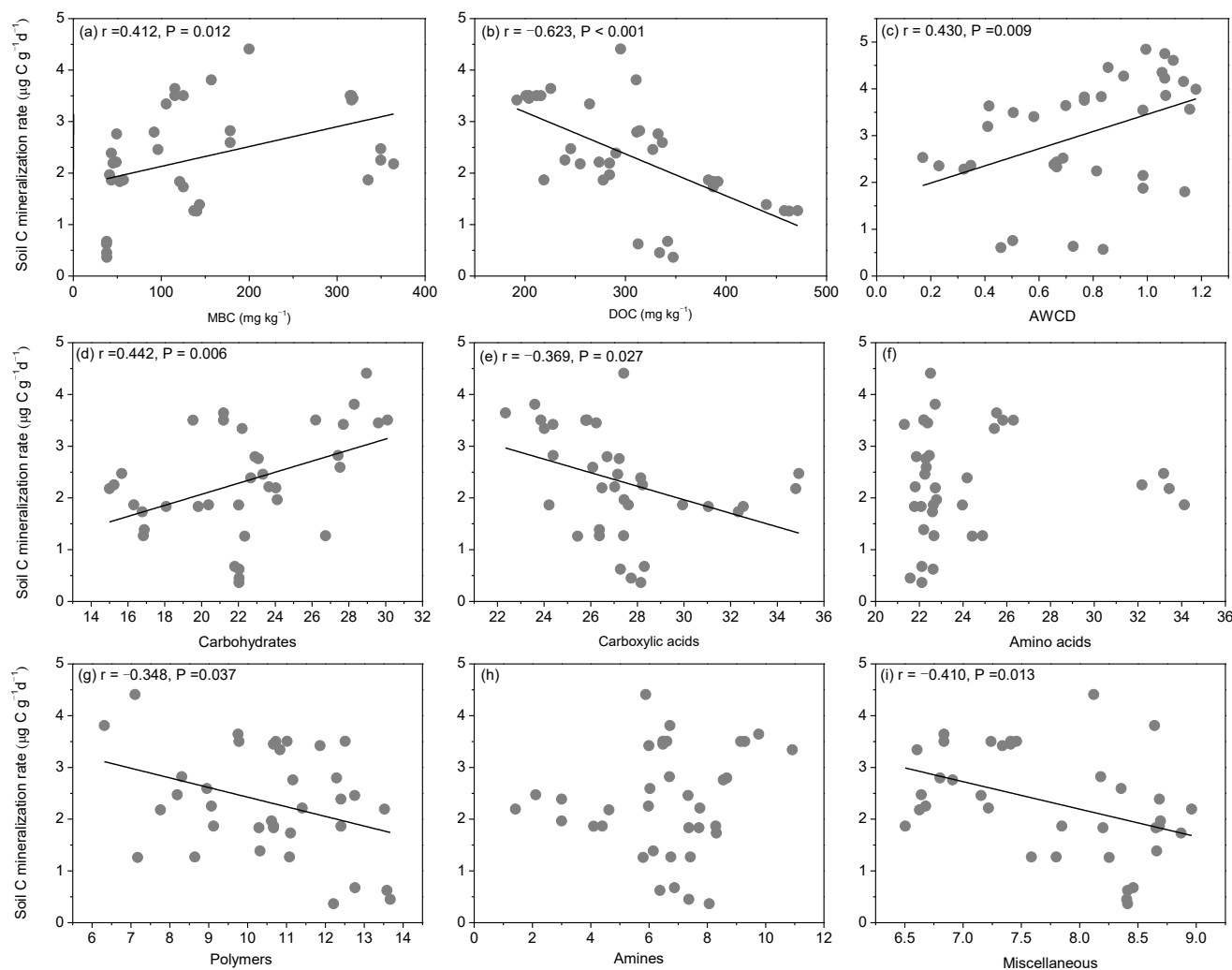

**Figure 6.** Relationships between soil carbon mineralization rates ($\mu g\ C\ g^{-1}\ d^{-1}$) and microbial biomass C (MBC), soil dissolved organic C (DOC), average well color development (AWCD), and
use of individual substrates.

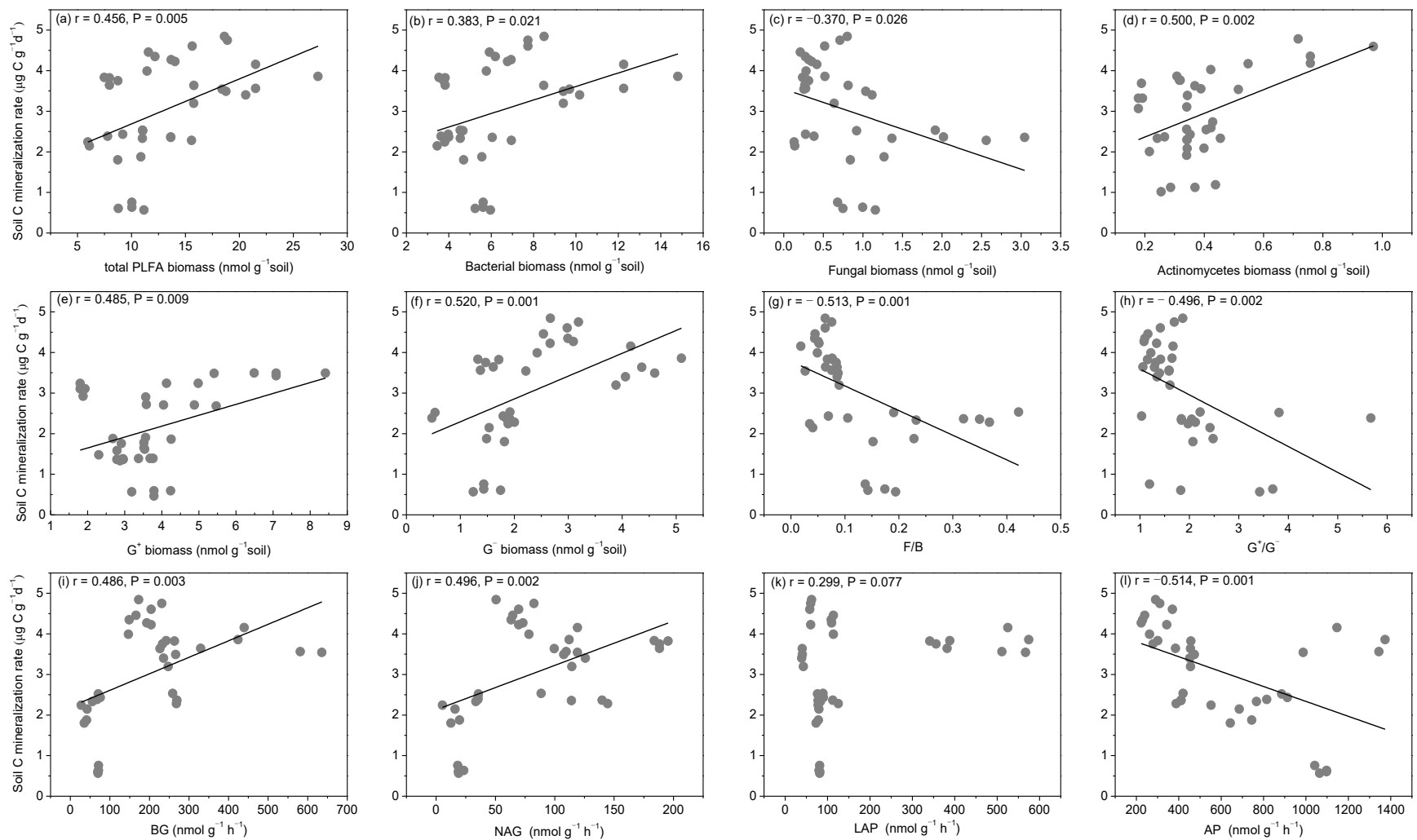

**Figure 7.** Relationships between soil carbon mineralization rates (μg C g$^{-1}$ d$^{-1}$) and different groups of soil microbial PLFAs (a-h) and enzyme activities (i-l).