# Peer review of "Plant functional traits determine latitudinal variations in soil microbial function: evidence from forests in"

_Biogeosciences, 2018_

## Referee Comment (RC1) · Anonymous Referee #2 · 21 Feb 2019

General comments

The Authors of the manuscript 'Plant functional traits determined the latitudinal variations in soil microbial functions: evidence from a forest transect in China' (bg-2018-499) by analyzing numerous parameters in forest soils located along North-South Transect of Eastern China (NSTEC) tried to answer the questions if: (1) the profiles of soil microbial substrate use varies along a latitudinal gradient, (2) biogeographical patterns of soil microbial substrate may be limited by climate and plant functional traits, and (3) the associations between soil microbial community and function could show functional dissimilarity. The Authors have found that soil microbial community structures

and functions were significantly correlated along NSTEC and that plant functional traits may influence patterns of soil microbial substrate. Moreover, based on analysis of relationships between soil microbial community structure and functions they concluded that there was functional dissimilarity.

In my opinion, the study is interesting and has merit; however it needs major revision. The methods have been properly designed and the results and reliable.

Specific comments

Major points:

The Authors must include section 'Chemicals' in which all compounds used in this study will be described including their name, purity/activity, place of purchase

The paper suffers from 'Abbreviations' section in which all important full and shortened names must be included. This will help the Authors to read the text.

Minor points:

Page 3, line 51, '….functional diversity to understand ..' functional diversity (of what?), please add Page 3, line 61-62, correct this sentence Page 3, line 73-74, correct to: 'reflect ….phosphorous and pesticides concentrations over . . .' Page 5, add section: 'Chemicals' Page 12, line 297, write 'positively' instead 'negatively' Page 13, line 336, correct to: 'species'

Technical comments

Write (for example) '20 âĄřC' instead of '20âĄřC' English of the paper should be corrected in some places

Please also note the supplement to this comment:
https://www.biogeosciences-discuss.net/bg-2018-499/bg-2018-499-RC1-supplement.pdf

---

## Short Comment (SC1) · 26 Feb 2019

Critical review: Plant functional traits determined the latitudinal variations in soil microbial functions: evidence from a forest transect in China

Gervolino, J., Irvine, S., Mokokwane, T., Saraogi, V., and Shearer, R. Summary:

The paper seeks to explore relationships between plant traits and microbial communities in soil. This is a pertinent question, especially in the context of ecological resilience and resistance. The main overall finding is that labile carbon is associated with microbial community composition, a clear but relatively unsurprising or limited conclusion.

[Figure]

There are some weaknesses in written presentation and in the presentation of data. The Abstract does not mirror the content of the main paper and lacks quantitative information. It is rather difficult to follow. In particular, the title does not reflect the real findings, as it is really a study of litter quality effects rather than plant functional traits.

In terms of format, the paper contains too many acronyms, which make the text hard to follow. Some of the acronyms not explained well enough. The text does flow well in many places and should be checked for readability.

The "community weighted mean" is central to the analysis, but the CMW abbreviation is not defined or discussed.

Specific points:

Introduction

The content lacks coherence and is occasionally repetitive. The text should have a more linear transition from plant to microbial function - and to isolate consideration of activity from diversity of community and community structure. The spatial dependence of microbial activity should be mentioned once at the outset, noting the issues of scales of spatial dependence.

The paper only briefly mentions plant functional traits as a determinant of ecosystem properties, especially for soil biogeochemical processes. The nature of the connection to microbial activity and function is poorly elucidated.

The introduction does not focus down to the study content until the end. It is difficult to understand the context of the study, since most of the introduction addresses how individual factors affects microbial activity individually.

Hypotheses are offered, but not in testable, directional form. They are broad and could be better stated as overarching questions considering how microbial substrates correlate with latitude as a reflection of litter quality / substate input.

[Figure]

In the last paragraph lines 112-116 consider the specific study sites / transect. This is contrived and should be raised in the Methods section, as the means by which the research question is addressed.

Methods

More detailed information is required concerning the sampling method. It is not adequate to refer only to a previously published article. It is not clear how the soil samples were analysed. It appears that there was no evaluation of the variability of the measured variables within the plots, as a basis for aggregation of the soils. The plots were quite large at 30 m x 40 m, without explanation for this. As such, the representivity of the bulked soil from mixing S-sampling is unknown

Results

The description of statistical analysis mentions correlation but in Fig. 6 and Fig. 7, the caption and associated text references relationships and the the plotted lines imply regression. The study is strictly correlative, since there are is no independent variable defined.

Discussion

This section is well organised by "theme" and well explained, each sub-section focused on one attribute of the results.

The introduction stated that study of relationships between soil microbial communities and microbial function are urgently needed. Here in line 382 the authors state that their findings the "functional dissimilarity hypothesis". This is useful, but line 382 also states that further studies are required to understand mechanisms that drive relationships between ecosystem functioning and soil microbial groups. Stating further research is needed is a poor way to end this section , without stating what exactly is next required. Was their experimental design suitable for studying the relationships in the first place?

Conclusion

The conclusion drawn is presumptuous on the limited evidence presented particularly with the lack of description for the term 'community weighted mean'.

---

## Referee Comment (RC2) · Anonymous Referee #3 · 28 Apr 2019

General comments

Relationship between plant functional traits and soil microbial functions is totally important research to estimate forest soil carbon and nutrient budget at present conditions and at the global climate change conditions. And meta-analysis using multi-site data or samples is one of the major methods to know it. However in this case, we need discreet data handling, appropriate hypothesis because each forest has specific and different conditions (e.g. plant, soil, environment, history) and interaction between functions and conditions is always complex In this MS, authors used 9 forests' soil samples and examined plant, carbon and microbe data. And authors defined this study as the re-

lation between "plant function and the latitudinal variations in soil microbial functions (title). And authors also mentioned that this study related with a counter-hypothesis about functional redundancy of microbe (L90-L111). However this MS has some unclear points in (1) hypothesis testing, (2) relation between plant and microbe and (3) latitudinal distribution. In my opinion, this research has much, reasonable and complex information however needs major revision.

Specific comments (1) Hypothesis testing At different forests and in different environmental conditions, specific (different) microbe distribution (species and activities) can happen and this must be common. Therefor in case mentioning on functional redundancy of microbe functions, careful definition of hypothesis is necessary because dissimilarity or similarity at multi sites does not directly mean functional redundancy of ecosystem. In the papers authors referred (L90-L111), Balser, Banerjee, Waldrop and Philippot used 1 site (or near 2 site transplanting) data and samples, and had a very clear hypothesis and testing. Strickland used several sites but experimental design was clear. In the study of Fierer, they used 71 site's samples but they focusing on bacteria (I recommend authors check this MS well.). Most of all studies conducted a specific manipulation and experiment for hypothesis testing because verification of functional redundancy in the steady state condition is difficult. On the other hand, I could not find one or several clear hypothesis in this paper. Please set more appropriate and clear hypothesis

(2) Relation between plant functional traits and soil microbial functions

In this MS, plant functional traits were defined in table S2 and used in Fig 4. Plant functional traits authors used were unclear in representativeness as plat functions. Because many researches focus on various plant functional traits which could regulate microbial activities and species distributions (e.g. priming effect, home field advantage, and fine root production). At least, I think authors need to mention why they choose these variables as plant functional traits. And several variables were discussed in section 4.2. Each relation was reasonable but not seemed to lead to functional redundancy

of microbe functions along forest sites

(3) Latitudinal distribution

This MS was defined as "the latitudinal variations in soil microbial functions". However I could not know about latitudinal distribution in soil microbial functions but comparison between forest and forest types. If authors wanted to assert this, I think they need focus more not on simple negative-positive relation but distribution (e.g. focusing on MAT vs plant type).

Technical comments

1. Scatter plots CSU vs leaf N (L298), CSU vs leaf C (L308) and LDMC(L321) may support readability. 2. Definition of CWM and H' was not clear.

---

## Author Comment (AC1) · 17 May 2019

Reviewer #1: General comments The Authors of the manuscript 'Plant functional traits determined the latitudinal variations in soil microbial functions: evidence from a forest transect in China' (bg-2018-499) by analyzing numerous parameters in forest soils located along North-South Transect of Eastern China (NSTEC) tried to answer the questions if: (1) the profiles of soil microbial substrate use varies along a latitudinal gradient, (2) biogeographical patterns of soil microbial substrate may be limited by climate and plant functional traits, and (3) the associations between soil microbial community and function could show functional dissimilarity. The Authors have found

that soil microbial community structures and functions were significantly correlated along NSTEC and that plant functional traits may influence patterns of soil microbial substrate. Moreover, based on analysis of relationships between soil microbial community structure and functions they concluded that there was functional dissimilarity. In my opinion, the study is interesting and has merit; however it needs major revision. The methods have been properly designed and the results and reliable. Specific comments Major points: (1) The Authors must include section 'Chemicals' in which all compounds used in this study will be described including their name, purity/activity, place of purchase Response: We have added the Chemicals used in this study as follow: The Biolog-ECO plates were purchased from Biolog, US. The substrates for BG, NAG, AP, and LAP were 4-MUB-$\beta$-D-glucoside, 4-MUB-N-acetyl-b-D-glucosaminide, 4-MUB-phosphate, and L-Leucine-7-amino-4-methylcoumarin, and were stored at $-20$ °C. An MUB standard was used for the BG, NAG, and AP enzymes and an AMC standard was used for the LAP enzyme. The substrates and standards were purchased Sigma. Analytical grade reagents were used for the soil nutrient analysis. (P8, Line 200-204) (2) The paper suffers from 'Abbreviations' section in which all important full and shortened names must be included. This will help the Authors to read the text. Response: we have added "Abbreviations" sections including all important full and shortened names as follow (P3, Line 54-81): Abbreviations: NSTEC North-South Transect of Eastern China AWCD Average well color development RDA Redundancy analysis Soil microbial community PLFAs Phospholipid fatty-acids G+ Gram positive bacteria G− Gram negative bacteria F/B Fungi/Bacteria Soil enzyme activities BG $\beta$-glucosidase NAG N-acetylglucosaminidase AP Acid phosphatase LAP Leucine aminopeptidase Soil properties SMC Soil moisture content SOM Soil organic matter SOC Soil organic carbon TN Total Nitrogen DOC Dissolved organic carbon MBC Microbial biomass carbon Silt Soil silt fractions (<53 $\mu$m) Plant functional properties: CWM Community-weighted means SLA The specific leaf area LDMC Leaf dry matter content Leaf C Leaf C concentrations Leaf N Leaf N concentrations Minor points: (3) Page 3, line 51, '….functional diversity to understand ..' functional

diversity (of what?), please add Response: we have rewritten this sentence as "It can be used to test fundamental questions about soil biological resistance and resilience (Jagadamma et al., 2014; Swallow and Quideau, 2015), and help us understand the role of microbial communities in different environments (Preston-Mafham et al., 2002)." (P4, Line 84-87) (4) Page 3, line 61-62, correct this sentence Response: we have rewritten this sentence as" For example, Tian et al. (2015), from their study of Changbai Mountain, China, found that the soil microbial metabolic activity and functional diversity were spatially dependent." (P4, Line 95-97). (5) Page 3, line 73-74, correct to: 'reflect ....phosphorous and pesticides concentrations over ...' Response: we have rewrite this sentence as "For example, the geographic patterns in soil microbial activities mainly reflect the climate, soil pH, and total phosphorus concentrations over large geographic scales (Cao et al., 2016)." (P4, Line 109-111). (6) Page 5, add section: 'Chemicals' Response: We have added the Chemicals used in this study (P8, Line 200-204). (7) Page 12, line 297, write 'positively' instead 'negatively' Response: The correlation coefficient in the Figure 4 refers to the correlations between the environmental properties and the RDA1 or RDA2. Silt and SMC were negatively correlated and Leaf N, Leaf C, and LDMC were significant positively correlated with RDA1, MAP, SOC, and TN were significant correlated with RDA2. The RDA1 represent the major variance of soil microbial carbon use efficiency. (8) Page 13, line 336, correct to: 'species' Response: DONE (P16, Line 438). Technical comments (9) Write (for example) '20 âĄřC' instead of '20âĄřC'ãĂĆ Response: DONE (P7, Line 178). (10) English of the paper should be corrected in some places Response: According to the comments and suggestions, we have carefully revised our manuscript. We rewritten the long sentences all through the text. In addition, we have our revised version manuscript professionally edited by a native English speaker colleague, Dr Deborah Ballantine from the United International College, Beijing Normal University-Hong Kong Baptist University.

Please also note the supplement to this comment:

https://www.biogeosciences-discuss.net/bg-2018-499/bg-2018-499-AC1-supplement.pdf

[Figure]

**Supplement:**

[revised manuscript text omitted]

---

## Author Comment (AC2) · 17 May 2019

Reviewer #2 General comments Relationship between plant functional traits and soil microbial functions is totally important research to estimate forest soil carbon and nutrient budget at present conditions and at the global climate change conditions. And meta-analysis using multi-site data or samples is one of the major methods to know it. However in this case, we need discreet data handling, appropriate hypothesis because each forest has specific and different conditions (e.g. plant, soil, environment, history) and interaction between functions and conditions is always complex In this MS, authors used 9 forests' soil samples and examined plant, carbon and microbe

data. And authors defined this study as the relation between "plant function and the latitudinal variations in soil microbial functions (title). And authors also mentioned that this study related with a counter-hypothesis about functional redundancy of microbe (L90-L111). However this MS has some unclear points in (1) hypothesis testing, (2) relation between plant and microbe and (3) latitudinal distribution. In my opinion, this research has much, reasonable and complex information however needs major revision. Specific comments (11) Hypothesis testing. At different forests and in different environmental conditions, specific (different) microbe distribution (species and activities) can happen and this must be common. Therefore in case mentioning on functional redundancy of microbe functions, careful definition of hypothesis is necessary because dissimilarity or similarity at multi sites does not directly mean functional redundancy of ecosystem. In the papers authors referred (L90-L111), Balser, Banerjee, Waldrop and Philippot used 1 site (or near 2 site transplanting) data and samples, and had a very clear hypothesis and testing. Strickland used several sites but experimental design was clear. In the study of Fierer, they used 71 site's samples but they focusing on bacteria (I recommend authors check this MS well.). Most of all studies conducted a specific manipulation and experiment for hypothesis testing because verification of functional redundancy in the steady state condition is difficult. On the other hand, I could not find one or several clear hypothesis in this paper. Please set more appropriate and clear hypothesis Response: we have rewritten the hypothesis part as "We tested four hypotheses in this study, as follows: (1) The profiles of soil microbial substrate use vary along a latitudinal gradient, (2) the functional characteristics of soil microbes are similar in closely related forest types, (3) biogeographical patterns of soil microbial substrate use are constrained by climate and plant functional traits, and (4) different soil microbial communities may have substrate use profiles and SOM decomposition rates." (P6, Line 160-164) (12) Relation between plant functional traits and soil microbial functions In this MS, plant functional traits were defined in table S2 and used in Fig 4. Plant functional traits authors used were unclear in representativeness as plant functions. Because many researches focus

on various plant functional traits which could regulate microbial activities and species distributions (e.g. priming effect, home field advantage, and fine root production). At least, I think authors need to mention why they choose these variables as plant functional traits. And several variables were discussed in section. Response: we have added the reason for the variables selection in the discussion as "A growing number of studies reported that vegetation type, land use, soil nutrients, and soil organic matter quality and quantity can determine large scale patterns of microbial communities (de Vries et al., 2012; Tu et al., 2016). Plant functional traits that are related to growth may determine a tree species' ability to contribute to the soil carbon pool via leaf litter inputs. For example, it was previously reported that plant traits such as the leaf N content, SLA, and LDMC could explain variations in soil nutrients and litter decomposition rates (Eichenberg et al., 2014; Laughlin, 2011). Therefore, we examined how these plant traits influenced the soil microbial function by latitude." (P14, Line 358-364) (13) 4.2. Each relation was reasonable but not seemed to lead to functional redundancy of microbe functions along forest sites Response: We have rewritten this part as "The functional dissimilarity of microbes and fungi may help explain these results. However, we did not measure some key variables, such as the microbial competition and interactions, and relationship between the microbial diversity and the decomposition rates. Therefore, in the future, we will use different experimental techniques that will help us gain an improved understanding of the mechanisms that drive the relationships between the structure and function of microbial communities." (P17, Line 450-455) (14) Latitudinal distribution This MS was defined as "the latitudinal variations in soil microbial functions". However I could not know about latitudinal distribution in soil microbial functions but comparison between forest and forest types. If authors wanted to assert this, I think they need focus more not on simple negative-positive relation but distribution (e.g. focusing on MAT vs plant type). Response: we have added the description on the variation of soil microbial functions along the latitude and between forest types in 3.1 section as follow: "The substrate microbial use ability was highest in the coniferous broad-leaved mixed forest and tropical forest soils, and lowest in the

coniferous forest soil (Fig. 3)." (P11, Line 280-282)." Of the six groups of C substrates, microbial communities in the temperate forests mainly used carbohydrates, carboxylic acids, and amino acids, which suggests that microorganisms in temperate forests probably use high-energy substrates that degrade easily (Kunito et al., 2009). The carbon substrate use was lowest in the coniferous forest. This shows that, compared with coniferous species, broadleaved tree species produce root exudates and litter high in water-soluble sugars, organic acids, and amino acids that are more favorable for microbial activity (Priha et al. 2001). There was no significant latitudinal pattern in the C metabolic intensity of soil microbes in our study, which was inconsistent with hypothesis (1). Our results show that MAP only had a moderate effect on the soil microbial function (Fig. 4). However, there was significant spatial variation in the use of different carbon sources, which was also related, to a lesser extent, to climate. Consistent with hypothesis (2), soil microbial functions were similar in closely related tree species and diverged as the variability between tree species and forest types increased (Fig. 4), which suggests that plant traits have more influence on soil microbial functions than climate. (P13, Line 347-357) We have also realized that the topic of 4.2 section was not conform with its text, so we have written it as "Mechanisms driving latitudinal variations in microbial substrate use" (P13, Line 343) Technical comments (15) Scatter plots CSU vs leaf N (L298), CSU vs leaf C (L308) and LDMC(L321) may support readability. Response: we have added the scatter plots about CSU and leaf N, leaf C and LDMC (Figure S2). (16) Definition of CWM and H' was not clear. Response: We have added the definition of the CWM and H' as follows: We established four sampling plots (30×40 m) in each forest ecosystem. In each plot, we recorded all the tree individuals, and measured the height and diameter-at-breast-height (DBH) of each woody individual with a DBH≥2 cm. The diversity of the tree species in the sampling plots was represented by $H\hat{E}z$, and the diversity ($H\hat{a}\check{A}\check{s}$, Shannon-Wiener) of the tree species in the community was calculated as follows: $H'=\sum\_(i = 0)\Theta_i(Pi \ln Pi) Where Pi was the importance value of the species i as a proportion of all species, and n was the number of the species.($ trait_i Where pi is the relative contribution of the species i to the cover of the whole

community, n is the number of the most abundant species, and trait i is the trait value of species i, as described by Garnier et al. (2004). The diversity of the tree species and plant functional traits are summarized in Table S2. (P8, Line 221-226)

Please also note the supplement to this comment:
https://www.biogeosciences-discuss.net/bg-2018-499/bg-2018-499-AC2-supplement.pdf

―――――――――――――――――――

---

## Author Comment (AC3) · 17 May 2019

(17) The paper seeks to explore relationships between plant traits and microbial communities in soil. This is a pertinent question, especially in the context of ecological resilience and resistance. The main overall finding is that labile carbon is associated with microbial community composition, a clear but relatively unsurprising or limited conclusion. There are some weaknesses in written presentation and in the presentation of data. The Abstract does not mirror the content of the main paper and lacks quantitative information. It is rather difficult to follow. In particular, the title does not reflect the real findings, as it is really a study of litter quality effects rather than

plant functional traits. Response: We have rewritten the abstract and added some quantitative information (P2, Line 35-41; Line 44-49; Line 51-53). We have add some detailed information about the latitudinal pattern of soil microbial carbon substrate use (P10, Line 280-282) and pertinent discussion (P13, Line 347-357). In order to explain the effect of plant traits on soil microbial function, we have added the scatter plots of the plant functional traits and carbon substrates use (Figure S2, supporting information). In our study, we did not directly analysis the influence of the litter quantity and quality on soil microbial function. However, we have added discussion about the influence of plant functional traits on litter (P14, Line 358-364). (18) In terms of format, the paper contains too many acronyms, which make the text hard to follow. Some of the acronyms not explained well enough. The text does flow well in many places and should be checked for readability. The "community weighted mean" is central to the analysis, but the CWM abbreviation is not defined or discussed. Response: We have defined the CWM abbreviation in our manuscript: "We also calculated the community-weighted means (CWM) values of the tree traits using the cover of each tree." (P8, Line 214-215) "To measure the leaf traits at the community level, we calculated the CWM of the tree layer, as follows: CWM=$\sum\_(i = 1)^n pi\times$ trait_i Where pi is the relative contribution of the species i to the cover of the whole community, n is the number of the most abundant species, and trait i is the trait value of species i, as described by Garnier et al. (2004). The diversity of the tree species and plant functional traits are summarized in Table S2." (P9, Line 221-226). In section 4.2 of discussion, we mainly discussed the effect of CWM of LDMC, leaf C, and leaf N on soil microbial carbon source use. We have added "Abbreviations" sections including all important full and shortened names as follow (P3, Line 54-81): Abbreviations: NSTEC North-South Transect of Eastern China AWCD Average well color development RDA Redundancy analysis Soil microbial community PLFAs Phospholipid fatty-acids G+ Gram positive bacteria G− Gram negative bacteria F/B Fungi/Bacteria Soil enzyme activities BG $\beta$-glucosidase NAG N-acetylglucosaminidase AP Acid phosphatase LAP Leucine aminopeptidase Soil properties SMC Soil moisture content SOM Soil

organic matter SOC Soil organic carbon TN Total Nitrogen DOC Dissolved organic carbon MBC Microbial biomass carbon Silt Soil silt fractions (<53 $\mu$m) Plant functional properties: CWM Community-weighted means SLA The specific leaf area LDMC Leaf dry matter content Leaf C Leaf C concentrations Leaf N Leaf N concentrations (19) The content lacks coherence and is occasionally repetitive. The text should have a more linear transition from plant to microbial function - and to isolate consideration of activity from diversity of community and community structure. The spatial dependence of microbial activity should be mentioned once at the outset, noting the issues of scales of spatial dependence. Response: we have carefully read our manuscript again and deleted that repeated part all through the text especially in section 4.3. (P16, Line 409-455) We mainly discussed the effect of plant functional traits on soil microbial function on section 4.2. (P14, Line 374-398) In addition, we added the spatial dependence of microbial activities in section 4.2 as "Of the six groups of C substrates, microbial communities in the temperate forests mainly used carbohydrates, carboxylic acids, and amino acids, which suggests that microorganisms in temperate forests probably use high-energy substrates that degrade easily (Kunito et al., 2009). The carbon substrate use was lowest in the coniferous forest. This shows that, compared with coniferous species, broadleaved tree species produce root exudates and litter high in water-soluble sugars, organic acids, and amino acids that are more favourable for microbial activity (Priha et al. 2001). There was no significant latitudinal pattern in the C metabolic intensity of soil microbes in our study, which was inconsistent with hypothesis (1). Our results show that MAP only had a moderate effect on the soil microbial function (Fig. 4). However, there was significant spatial variation in the use of different carbon sources, which was also related, to a lesser extent, to climate. Consistent with hypothesis (2), soil microbial functions were similar in closely related tree species and diverged as the variability between tree species and forest types increased (Fig. 4), which suggests that plant traits have more influence on soil microbial functions than climate." (P13, Line 347-357) (20) The paper only briefly mentions plant functional traits as a determinant of ecosystem properties, especially

for soil biogeochemical processes. The nature of the connection to microbial activity and function is poorly elucidated. Response: We have discussed connection to microbial community and soil carbon substrate use, enzyme activities, and SOM decomposition rate as in section 4.3 (P16, Line 419-436). (21) The introduction does not focus down to the study content until the end. It is difficult to understand the context of the study, since most of the introduction addresses how individual factors affects microbial activity individually. Response: in the second paragraph, we focus on the spatial pattern of soil microbial communities, enzyme activities, and metabolic activities in different scales. However, there was no studies about the variation of the microbial substrate use in large scale which support our hypotheses (1). (P4, Line 91-106) In the third paragraph we focus on the environmental properties which influence the soil microbial communities and activities. However, we still don't know about climate and plant functional traits which one is more important for the variation in soil microbial substrate use and this support our hypotheses (2) and hypotheses (3). (P4, Line 107-125) In the fourth paragraph, we focus on the relationship between soil microbial communities and function (hypotheses (4)). (P5, Line 126-148) (22) Hypotheses are offered, but not in testable, directional form. They are broad and could be better stated as overarching questions considering how microbial substrates correlate with latitude as a reflection of litter quality / substrate input. Response: We have rewritten our Hypotheses as "We tested four hypotheses in this study, as follows: (1) The profiles of soil microbial substrate use vary along a latitudinal gradient, (2) the functional characteristics of soil microbes are similar in closely related forest types, (3) biogeographical patterns of soil microbial substrate use are constrained by climate and plant functional traits, and (4) different soil microbial communities may have substrate use profiles and SOM decomposition rates." (P6, Line 160-164).

Please also note the supplement to this comment:
https://www.biogeosciences-discuss.net/bg-2018-499/bg-2018-499-AC3-supplement.pdf

---

## Author Comment (AC4) · 17 May 2019

Reviewer #1: General comments The Authors of the manuscript 'Plant functional traits determined the latitudinal variations in soil microbial functions: evidence from a forest transect in China' (bg-2018-499) by analyzing numerous parameters in forest soils located along North-South Transect of Eastern China (NSTEC) tried to answer the questions if: (1) the profiles of soil microbial substrate use varies along a latitudinal gradient, (2) biogeographical patterns of soil microbial substrate may be limited by climate and plant functional traits, and (3) the associations between soil microbial community and function could show functional dissimilarity. The Authors have found

that soil microbial community structures and functions were significantly correlated along NSTEC and that plant functional traits may influence patterns of soil microbial substrate. Moreover, based on analysis of relationships between soil microbial community structure and functions they concluded that there was functional dissimilarity. In my opinion, the study is interesting and has merit; however it needs major revision. The methods have been properly designed and the results and reliable. Specific comments Major points: (1) The Authors must include section 'Chemicals' in which all compounds used in this study will be described including their name, purity/activity, place of purchase Response: We have added the Chemicals used in this study as follow: The Biolog-ECO plates were purchased from Biolog Company in America. The substrates for BG, NAG, AP, and LAP were 4-MUB-$\beta$-D-glucoside, 4-MUB-N-acetyl-b-D-glucosaminide, 4-MUB-phosphate, and L-Leucine-7-amino-4-methylcoumarin, and were stored at -20âĐČ. The standard liquid for BG, NAG, and AP enzymes was MUB and AMC for LAP enzyme. All substrates and standard liquid were purchased from Sigma Company. The reagents used for soil nutrients were the analytical reagents. (P8, Line 200-205) (2) The paper suffers from 'Abbreviations' section in which all important full and shortened names must be included. This will help the Authors to read the text. Response: we have added "Abbreviations" sections including all important full and shortened names as follow (P3, Line 53-80): Abbreviations: NSTEC North-South Transect of Eastern China AWCD Average well color development RDA Redundancy analysis Soil microbial community PLFAs Phospholipid fatty-acids G+ Gram positive bacteria G− Gram negative bacteria F/B Fungi/Bacteria Soil enzyme activities BG $\beta$-glucosidase NAG N-acetylglucosaminidase AP Acid phosphatase LAP Leucine aminopeptidase Soil properties SMC Soil moisture content SOM Soil organic matter SOC Soil organic carbon TN Total Nitrogen DOC Dissolved organic carbon MBC Microbial biomass carbon Silt Soil silt fractions (<53 $\mu$m) Plant functional properties: CWM Community-weighted means SLA The specific leaf area LDMC Leaf dry matter content Leaf C Leaf C concentrations Leaf N Leaf N concentrations Minor points: (3) Page 3, line 51, '….functional diversity to understand ..' functional diversity

(of what?), please add Response: we have rewritten this sentence as "We need robust information about soil microbial functional diversity to understand the role of microbial communities in different environments (Preston-Mafham et al., 2002)" (P4, Line 84-86) (4) Page 3, line 61-62, correct this sentence Response: we have rewritten this sentence as" For example, Tian et al. (2015) have found that the soil microbial metabolic activity and functional diversity were spatially dependent in their study of Changbai Mountain in China." (P4, Line 95-97). (5) Page 3, line 73-74, correct to: 'reflect . . ..phosphorous and pesticides concentrations over . . .' Response: we have rewrite this sentence as "For example, the geographic patterns of soil microbial activity are mainly reflected by the climate, soil pH, and total phosphorus concentrations over large spatial scales (Cao et al., 2016)". We have read this paper again and the authors have not refer to the word "pesticides concentrations". (P4, Line 108-110). (6) Page 5, add section: 'Chemicals' Response: We have added the Chemicals used in this study (P8, Line 200-205). (7) Page 12, line 297, write 'positively' instead 'negatively' Response: The correlation coefficient in the Figure 4 refers to the correlations between the environmental properties and the RDA1 or RDA2. Silt and SMC were negatively correlated and Leaf N, Leaf C, and LDMC were significant positively correlated with RDA1, MAP, SOC, and TN were significant correlated with RDA2. The RDA1 represent the major variance of soil microbial carbon use efficiency. (8) Page 13, line 336, correct to: 'species' Response: DONE (P16, Line 438). Technical comments (9) Write (for example) '20 âĄřC' instead of '20âĄřC'ãĂĆ Response: DONE (P7, Line 178-179). (10) English of the paper should be corrected in some places Response: According to the comments and suggestions, we have carefully revised our manuscript. We rewritten the long sentences all through the text. In addition, we have our revised version manuscript professionally edited by a native English speaker colleague, Dr Deborah Ballantine from the United International College, Beijing Normal University-Hong Kong Baptist University.

Please also note the supplement to this comment:

https://www.biogeosciences-discuss.net/bg-2018-499/bg-2018-499-AC4-supplement.pdf

---

## Author Comment (AC6) · 17 May 2019

The comment was uploaded in the form of a supplement:
https://www.biogeosciences-discuss.net/bg-2018-499/bg-2018-499-AC6-supplement.pdf

---

## Author Response (AR2)

Manuscript bg-2018-499-R2                    Response to Reviewers

Dear Editor,

We deeply appreciate you for giving us an opportunity to improve our manuscript again. We would like to thank all of you and the two reviewers for the thoughtful and valuable suggestions on our manuscript entitled "**Plant functional traits determined the latitudinal variations in soil microbial functions: evidence from a forest transect in China**" [ID: bg-2018-499]. According to the comments and suggestions, we have carefully revised our manuscript. We have followed the formatting requirements as presented in the Guide for Authors. We have uploaded the document of the responses to reviewer and a clean manuscript. Here are the point-to-point responses (color-coded blue) to Editor's and **Reviewer**'s comments (color-coded black). The page and line numbers mentioned here refer to our latest revision of our manuscript simultaneously submitted.

Reviewer #1:

Authors significantly improved this manuscript using reviewers' comments to clarify four hypotheses that motivate the work and explain numerous abbreviations. The large number of measurements made on each forest soil sample provides a great deal of correlative insight into factors that affect respiration potential. A few questions remain, which I expect the authors can address through clarification or some minor recalculations.

1) The Biolog and respiration rate experiments measured only aerobic activities. While these may be the most significant contributors to C mineralization, the authors should clarify that anaerobic microbial decomposition mechanisms may be important for intermittently saturated soils (such as CB at the time of sampling – Table 2) and sites with high mean annual precipitation. Such activities may have different effects on organic acid and DOC concentrations.

Response: The SOM decomposition and Biolog analysis were conducted in 2013-2014. We have no samples now to conduct the experiment about anaerobic microbial decomposition. We have added discussion about the anaerobic microbial decomposition mechanisms as "Some forest soils were intermittently saturated (such as CB, Table 2) or high with mean annual precipitation. Under the anaerobic conditions, soil organic decomposition is mediated by a complex suite of microbial processes (Megonigal et al., 2004). The fermentation products including low molecular weight alcohols, fatty acids, and dihydrogen can serve as substrates for anaerobic respiration using a variety of alternative terminal electron acceptors in place of oxygen to mineralize organic carbon to carbon dioxide ($CO_2$). Therefore, the soil organic matter decomposition rate might be slow in these anaerobic conditions. These results demonstrate that the reduction of organic matter is a key step of anaerobic decomposition (Keller and Takagi, 2013)." (P13-14, Line 356-364)

2) Were the significance tests of Pearson correlation coefficients (Figs 5-7 & S2) corrected for multiple hypothesis testing? Please explain in methods.

Response: All P values were adjusted using the Bonferroni correction to account for multiple comparisons. We have added the explanation in the methods. (P10, Line 266-267)

3) As explained in the cited Biolog methods paper by Guckert et al. Biolog results must be corrected for inoculum size. Others have used Average Well Color Development (AWCD) to normalize values, but that does not appear to be done here. Did the authors normalize values before integration for Figure 3 using microbial biomass or another indicator of inoculum size?

Response: To minimize the influence of cell density in comparisons among samples, results can be analyzed at constant average well color development (AWCD). The AWCD for each microplate was calculated by subtracting the control well optical density (OD) from the substrate well OD (blanked substrate wells), setting any resultant blanked substrate wells with negative values to 0 and taking the mean of the 95 blanked substrate wells.(P9,Line 238-242)

General comments

1. In previous comment I mentioned that "Please set more appropriate and clearer hypothesis". And in this version, numbers of hypothesis increased from 3 to 4. I think multiple hypotheses might cause complex or complicated discussion and conclusion even in case two. In this version, authors added "information about the relationships between soil microbial communities and their functions in natural ecosystems is urgently needed" as necessity of this research after presenting 2 conflicting hypotheses. I agree with the opinion of field data-oriented validation's necessity. But simple hypothesis or aim to conduct 4 testing is required. Moreover, in case conducting 4 testing, relation between these testing must be described. For example, in section 4.2, hypothesis 1 seemed to be described as counter hypothesis of 2. And these hypotheses should be integrated in conclusion. I could not find any comment about latitudinal variations in conclusion.

Response: we have rewritten our hypotheses as "(1) The profiles of soil microbial substrate use vary along a latitudinal gradient, (2) biogeographical patterns of soil microbial substrate use are constrained by climate and plant functional traits, and (3) different soil microbial communities may have substrate use profiles and SOM decomposition rates." (P6, Line 165-168)

We have rewritten this part as "There was no significant latitudinal pattern in the soil total microbial metabolic intensity (S) in our study, which was inconsistent with hypothesis (1). However, there was significant latitudinal variation in the use of the six different carbon sources. The soil microbial use of carbon sources were relatively higher in tropical and subtropical climatic area than those in temperate climatic area." (P14, Line 374-377)

We have added description on the latitudinal pattern of soil microbial use of different carbon sources in the conclusion part as "There was no obvious latitudinal pattern for the soil total microbial metabolic intensity. However, soil microbial use of the different carbon sources varied with the latitude except the amines carbon source. Soil microbial use of carbon sources were relatively higher in tropical climatic areas." (P18, Line 491-494)

2. Title of this MS include "latitudinal variations". And I mentioned as "I could not know about latitudinal distribution in soil microbial functions but comparison between forest and forest types". In this version authors mentioned that "plant traits have more influence on soil microbial functions than climate" and if it is correct, I think title of this MS can be "Plant functional traits determine soil microbial function". However, in this case, careful discussion must be needed. Does only MAT represent variation of climate? How do authors count indirect effect of climate (at least climate condition regulate forest type)? Etc. For example, in the paper of Tu(2016), their conclusion was not "climate is not main regulator" but "This study suggests that even microbial subcommunities follow general biogeographic patterns". And they used several indexes reflecting latitudinal gradient.

Response: we have added the description on the latitudinal pattern of soil microbial functions as "There was no significant latitudinal pattern in the soil total microbial metabolic intensity in our study, which was inconsistent with hypothesis (1). However, there was significant latitudinal variation in the use of different carbon sources. The soil microbial use of carbon sources were relatively higher in tropical and subtropical climatic area than those in temperate climatic area." (P14, Line 374-377)

We have conducted the path analysis relevant environmental variables to soil microbial use of different carbon sources (Table S4-9, supporting information). We found that the MAT and MAP had indirect effect on the soil microbial use of different carbon sources through influencing the soil temperature, soil total nitrogen, and total phosphorus. (P14, Line 381-383)

We have written this part as "Consistent with hypothesis (2), soil microbial functions were similar in closely related to the CWM values of the tree species (Fig. 4). However, only MAP of climatic factors had a moderate effect on the soil microbial function (Fig. 4). Climate may have indirect effect on the latitudinal pattern of different carbon sources use. In our study, MAT and MAP affected the six different carbon sources use indirectly by influencing the soil temperature (ST), soil total nitrogen (TN), and total phosphorus (TP) (Table S4-9). It was reported that climate was the main environmental parameters driving the latitudinal patterns of community-level leaf traits through the regulation of species composition (Wang et al., 2016). Woody plant species with evergreen broadleaves dominated in the tropic regions characterized by hot, humid, and infertile habitats showed low CWM values of leaf N and SLA and high CWM values of LDMC (Table S2)." (P14, Line 378-387)

Technical comments

1. (P2, L36 37 38) Meaning of intensities are not clear.

Response: They means the soil microbial substrate use intensities. We have added the full meanings of them in the text. (P2, Line 36-38)

2. (P2, L39) What index do authors use as Plant functional traits and climate?

Response: the indexes included in the plant functional traits were leaf dry matter content, leaf C concentrations, and leaf N concentrations. The climate index was the mean annual precipitation. We have added them in the abstract. (P2, Line 39-41)

3. (P3) Abbreviations needs climate conditions.

Response: Done (P3, Line 65-67).

4. (P17, L462) Contribution of silt content is totally common.

Response: we have rewritten this part as "For the first time, we showed that CWM values of plant traits explained variations in soil microbial C substrate use at the regional scale. Additionally, the fine-grained soils with high silt contents were higher in the microbial C substrate use." (P18, Line 494-498)

Thanks again to you and the two reviewers for the thoughtful and thorough comments.

We hope that our revisions will be satisfactory, and we are very happy to work with you and the reviewers to resolve any remaining problems.

Yours sincerely,

Zhiwei Xu, Guirui Yu, Qiufeng Wang, Xinyu Zhang, Ruili Wang, Ning Zhao, Nianpeng He, Ziping Liu

A list of all relevant changes

1. Affiliations of the first authors (P1,line7-12)

2. Abstract: (P2,line36-41)

3. Abbreviations (P3,line65-67)

4. Introduction   (P6,line165-168)

5. Material and methods(2.4(P9,line238-242); 2.6(P10,line266-267; line275-278)

6. Result (P11,line281,287-289)

7. Discussion (4.1 (P13-14,line356-364); 4.2(P14,line374-387)

8. Conclusion (P18,line 491-498)

9. References (P20,line 574-575; 594-595; P21,line 673-674)

[revised manuscript text omitted]